# DHX36 prevents the accumulation of translationally inactive mRNAs with G4-structures in untranslated regions

Markus Sauer[1,2,3], Stefan A. Juranek[3], James Marks[4], Alessio De Magis[3], Hinke G. Kazemier[2], Daniel Hilbig[3], Daniel Benhalevy [4], Xiantao Wang[4], Markus Hafner [4] & Katrin Paeschke [1,2,3]

Translation efficiency can be affected by mRNA stability and secondary structures, including G-quadruplex structures (G4s). The highly conserved DEAH-box helicase DHX36/RHAU resolves G4s on DNA and RNA in vitro, however a systems-wide analysis of DHX36 targets and function is lacking. We map globally DHX36 binding to RNA in human cell lines and find it preferentially interacting with G-rich and G4-forming sequences on more than 4500 mRNAs. While DHX36 knockout (KO) results in a significant increase in target mRNA abundance, ribosome occupancy and protein output from these targets decrease, suggesting that they were rendered translationally incompetent. Considering that DHX36 targets, harboring G4s, preferentially localize in stress granules, and that DHX36 KO results in increased SG formation and protein kinase R (PKR/EIF2AK2) phosphorylation, we speculate that DHX36 is involved in resolution of rG4 induced cellular stress.

[1] Department of Biochemistry, Biocenter, University of Würzburg, 97074 Würzburg, Germany. [2] European Research Institute for the Biology of Ageing (ERIBA), University Medical Center Groningen, University of Groningen, 9713 AV Groningen, The Netherlands. [3] Department of Oncology, Hematology and Rheumatology, University Hospital Bonn, 53127 Bonn, Germany. [4] Laboratory of Muscle Stem Cells and Gene Regulation, National Institute of Arthritis and Musculoskeletal and Skin Diseases, NIH, Bethesda, MD 20892, USA. Correspondence and requests for materials should be addressed to M.H. (email: markus.hafner@nih.gov) or to K.P. (email: katrin.paeschke@ukbonn.de)

RNA can adopt a variety of structures that have important roles for its function and stability[1]. Among the most stable nucleic acid structures are G-quadruplexes (G4s), which at their core contain stacked guanine tetrads built by Hoogsteen hydrogen bonding[2]. In recent years, intense efforts were directed towards identifying RNA G4 structures (rG4s) and understanding their impact on gene regulation in normal and disease cells[3–7]. In vitro and in silico approaches have revealed over 13,000 sites in the human transcriptome with the potential to form rG4s[8–10], further supported by immunofluorescence experiments with a specific antibody that detected rG4s in the cytoplasm of human cells[11]. These rG4s may influence many aspects of posttranscriptional regulation, including alternative polyadenylation, splicing, and miRNA biogenesis[3,7,12–15]. Best documented is their impact on translation, where rG4s in 5′ untranslated regions (UTR) and possibly 3′ UTRs of messenger RNA (mRNA) negatively affect translation[5,16,17]. Taken together these findings provide strong, albeit indirect, evidence supporting a model that rG4s form in vivo and have gene regulatory function.

Nevertheless, the existence and relevance of rG4s remain the subject of vigorous and controversial discussion[18]. Chemical mapping in mammalian cells suggested they are globally unfolded by the action of an uncharacterized but essential machinery[9]. Consequently, there is an urgent need to identify the components of this machinery and dissect their function, in order to understand the role of rG4s in RNA metabolism. The majority of candidates to resolve rG4s are ATP-dependent RNA helicases[19], but also include some sequence-specific RNA-binding proteins (RBPs)[20]. A large level of redundancy of the rG4-interacting machinery is expected, considering that neither transient knockdown of a helicase, DHX36, nor ATP depletion have resulted in the increase of rG4 formation above the threshold that could be reliably detected by dimethyl sulfate sequencing[9].

So far, only a handful of helicases are known to affect rG4 unfolding[19]. Among them is the 3′−5′ DEAH-box helicase DHX36, which has robust DNA and RNA G4 unwinding activity[21–23], but was initially reported to associate with AU-rich sequence elements and is also known as RHAU (RNA helicase associated with AU-rich element)[24]. Although a recent structural study[25] comprehensively illuminated the molecular mechanism of G4 unwinding by DHX36, its in vivo role was only studied using reporter gene constructs and a few individual target genes[21,26–29]. The full complement of DHX36 targets and its impact on posttranscriptional gene regulation remain unknown.

Here, we analyze DHX36 targets and its impact on gene regulation on a systems-wide scale. We identify DHX36 as a predominantly cytoplasmic RNA helicase that specifically interacts with G-rich sites of mRNAs previously shown to form rG4s in vitro[10]. Loss-of-function analysis with DHX36-KO cells coupled with RNA sequencing (RNA-seq), ribosome profiling (Ribo-seq), and high throughput proteomics show that binding of DHX36 in the 3′ and 5′ UTR results in higher target mRNA translational efficiency, in a helicase activity-dependent manner. Loss of DHX36 results in the accumulation of translationally incompetent target mRNAs. These mRNAs, harboring rG4s, preferentially localize in stress granules (SG). Furthermore, DHX36-KO increase SG formation and activates the protein kinase R (PKR/EIF2AK2)-mediated stress response. Taken together, we propose the model that DHX36 loss results in the formation of rG4s and other structures on target mRNAs that stabilize them, but also trigger a stress response rendering them translationally incompetent, possibly by sequestration in SGs or their precursors/seeds.

## Results
**DHX36 is a cytoplasmic helicase interacting with mRNA**. The molecular and structural basis for unwinding of G4 structures by DHX36 is well understood[25]. It is clear that the N-terminal domain together with an OB-fold of DHX36 specifically recognizes parallel DNA and RNA G4 and unfolds them in an ATP-dependent manner[25]. However, models for DHX36 in vivo function vary, including whether it preferentially acts as a DNA or an RNA helicase and whether it associates preferentially with G4 or AU-rich sequences. In order to dissect its cellular function, we first aimed to determine the subcellular localization of its two known splice isoforms that differ by alternative 5′ splice site usage in exon 13 (Supplementary Fig. 1a). We generated stable HEK293 cell lines expressing FLAG/HA-tagged DHX36 (FH-DHX36) isoforms 1 and 2 under control of a tetracycline-inducible promoter. Upon tetracycline induction, transgenic FH-DHX36 accumulated to approx. 4-fold higher levels compared to endogenous DHX36 in HEK293 cells (Fig. 1a). Using these cell lines, as well as parental HEK293 cells, we performed subcellular fractionation of DHX36 and found it predominantly localized to the cytoplasm in all cases (Fig. 1b). Because no changes in the two isoforms could be detected, we did not further discriminate between them.

Although DHX36 has been described as both a DNA and RNA helicase in vitro[23], the major source of nucleic acids in the cytoplasm is RNA. Thus, we tested in our transgenic HEK293 cells whether DHX36 interacted with RNA, in particular mRNAs, by purification of polyadenylated RNA after UV-crosslinking. We found that DHX36 was abundantly interacting with poly(A) RNA, indicating that cytoplasmic mRNAs are its main targets (Fig. 1c).

The interaction of DHX36 with mRNAs in the cytoplasm suggested a posttranscriptional regulatory function. Considering that DHX36 was previously proposed to function in translational regulation[29], we investigated whether DHX36 co-migrates with the translational machinery and fractionated HEK293 cell extracts by sucrose gradient ultracentrifugation. In proliferating cells, more than 90% of endogenous DHX36 were found in the soluble cytoplasm fractions and the remainder migrated with the monosomal and polysomal fractions (Fig. 1d). Changes in DHX36 expression, such as fourfold overexpression in our transgenic HEK293 cells does not alter this distribution (Supplementary Fig. 1b). Treatment of the cell extracts with RNase to collapse the polysomes led to a complete loss of DHX36 from the heavy fractions (Fig. 1d). Considering that the translation initiation factor EIF4A shows a similar distribution on polysomes, our data indicate that DHX36 does not affect translation elongation, but could possibly be involved in translation initiation.

**DHX36 binds thousands of sites on mature mRNAs**. In order to comprehensively capture DHX36 binding sites and characterize its RNA recognition elements (RREs), we mapped the RNA interactome of DHX36 in living cells on a transcriptome-wide scale at nucleotide (nt)-resolution using 4-thiouridine (4SU) PAR-CLIP[30]. UV-crosslinking of active helicases that rapidly translocate on their RNA targets may complicate identification of binding preferences due to transient and fast helicase progression. Therefore, we performed PAR-CLIP in two stable HEK293 cell lines, either inducibly expressing FH-DHX36 or the catalytically dead FH-DHX36-E335A mutant, which we expected to remain stuck at the sites of DHX36 action[22,25].

Autoradiography of the crosslinked, ribonuclease-treated, and radiolabeled FLAG-immunoprecipitate confirmed the isolation of one main ribonucleoprotein particle (RNP) at the expected size of ~116 kDa corresponding to the FH-DHX36 and FH-DHX36-E335A RNPs (Fig. 2a). We recovered bound RNA fragments from the RNPs of two biological replicates per cell line and

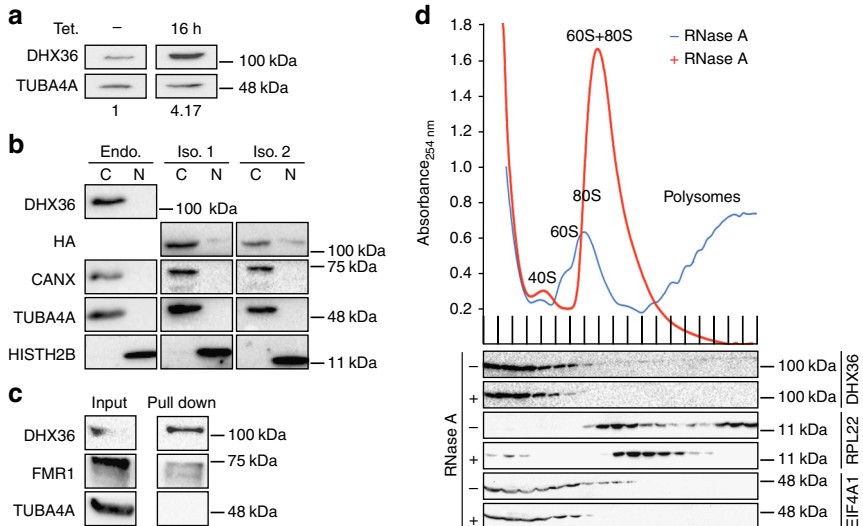

**Fig. 1** DHX36 is mainly cytoplasmic and does not directly interact with ribosomes. **a** Quantification of transgenic FLAG/HA(FH)-DHX36 expression upon 16 h induction with tetracycline (Tet.). TUBA4A-normalized DHX36 quantities are indicated below. **b** Endogenous (Endo.), as well as transgenic FH-tagged DHX36 isoform 1 (Iso. 1) and 2 (Iso. 2) show mainly cytoplasmic localization in biochemical fractionation experiments from HEK293 cells. Cytoplasmic (C) and nuclear (N) fractions were probed with anti-HISTH2B (nuclear marker), anti-TUBA4A (cytoplasmic marker), and anti-CANX (endoplasmic reticulum marker) antibodies. Endogenous DHX36 and transgenic FH-DHX36 isoforms 1 and 2 were detected with anti-DHX36 antibody and anti-HA antibody, respectively. **c** DHX36 can be co-purified with polyadenylated RNA. The RBP FMR1 served as a positive, TUBA4A as a negative control, respectively. **d** UV absorbances at 254 nm of RNase A-treated (red) and untreated (blue) HEK293 cell extracts separated by sucrose gradient centrifugation are shown. Peaks of UV absorbance corresponding to 40S, 60S, 80S ribosomes, and polysomes are indicated. Western blots probed for DHX36, RPL22, and EIF4A1 are shown below. Source data are provided as a Source Data file

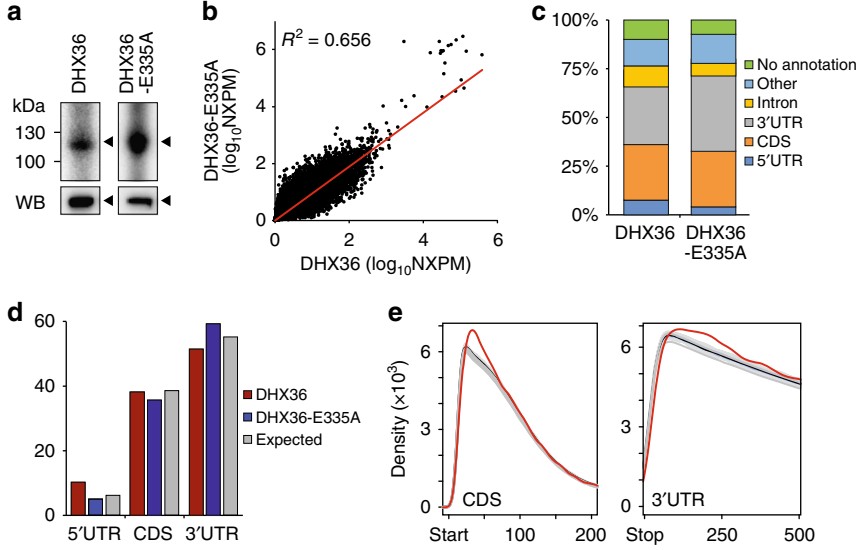

**Fig. 2** DHX36 interacts with mature mRNAs at thousands of sites. **a** Autoradiographs of crosslinked and radiolabelled FH-DHX36 and FH-DHX36-E335A RNPs separated by SDS-PAGE (black arrowheads). HA, immunoblot for haemagglutinin tag. Source data are provided as a Source Data file. **b** Scatterplot of normalized crosslinked reads per million (NXPM) from FH-DHX36 and FH-DHX36-E335A PAR-CLIP experiments reveals high degree of correlation of high-confidence binding sites. Correlation coefficient ($R^2$) is indicated. **c** Distribution of PAR-CLIP-derived binding sites from the intersection of two biological replicates across different RNA annotation categories. **d** The distribution of FH-DHX36 (red) and FH-DHX36-E335A (blue) binding sites across CDS, 3′, and 5′ UTRs matches the distribution expected based on the length of the annotation categories (gray). **e** Metagene analysis of the distribution of DHX36 binding clusters 200 nt downstream of the start codon and 500 nt downstream of the stop codon, respectively (red). The distribution of 1000 mismatched randomized controls is shown in gray. The black line indicates the mean of the gray distribution

transformed them into small RNA cDNA libraries for next-generation sequencing. Using the PARalyzer software[31], we determined clusters of overlapping reads that harbor characteristic T-to-C conversions diagnostic of 4SU-crosslinking events at higher frequencies than expected by chance (see Supplementary data 1 for summary statistics). The biological replicates showed

excellent correlation, with an $R^2$ of 0.79 and 0.93 for the FH-DHX36 and FH-DHX36-E335A PAR-CLIPs, respectively (Supplementary Fig. 2a, b) and allowed us to define a set of reproducible high-confidence binding sites of 19,585 and 67,660 clusters, respectively (Supplementary data 2). The binding profiles of FH-DHX36 and FH-DHX36-E335A were also highly

correlated, with an $R^2$ of 0.66, indicating that inactivation of the helicase domain did not interfere markedly with the binding pattern of the protein (Fig. 2b). Among our extensive target list were also 22 of 28 previously published targets of DHX36 (Supplementary data 3).

Consistent with its mainly cytoplasmic localization and interaction with polyadenylated mRNAs (Fig. 1b, c), 70% and 73% of FH-DHX36 and FH-DHX36-E335A binding sites, respectively, mapped to exons of more than 4500 different mRNAs (Fig. 2c). We did not observe any preference for FH-DHX36 and FH-DHX36-E335A binding sites to reside in coding sequence (CDS) or 3′ and 5′ UTR of mRNA targets compared to chance (Fig. 2d). Nevertheless, a metagene analysis revealed an enrichment of FH-DHX36-E335A binding sites within the first 100 nt of the start codon in the CDS, resembling the binding profile of another cytoplasmic G-rich binding protein, CNBP[20], as well as directly downstream of the stop codon (Fig. 2e and Supplementary Fig. 2c). Taken together, our data suggested that DHX36 bound and likely regulated a wide range of targets in the cytoplasm, possibly independent from the translation machinery.

**DHX36 binds G-rich targets in cells that form rG4s in vitro.** Binding of FH-DHX36 and FH-DHX36-E335A to mRNAs showed no correlation to transcript length or abundance as determined by RNA-seq in HEK293 cells (Supplementary Fig. 2d–g). This suggested sequence- or structure-dependent determinants of FH-DHX36 binding, rather than unspecific interactions. To determine the RRE of FH-DHX36 and FH-DHX36-E335A, the occurrence of all possible 5mer sequences in our high-confidence binding sites were counted and their Z-scores over a background of shuffled sequences of the same nucleotide composition were calculated. Both PAR-CLIP data sets showed an enrichment in 5mers that contain at least three guanines (Fig. 3a, Supplementary data 2), but, in the FH-DHX36-E335A PAR-CLIP additional A/U-rich 5mers were identified (Fig. 3a and Supplementary data 2). Similarly, MEME[32] revealed a G-rich RRE, that matched the criteria for rG4 formation[33] (Fig. 3b). Oligonucleotides corresponding to the consensus RRE or to PAR-CLIP binding sites of four top target genes indeed folded into G4s in vitro confirmed by circular dichroism spectrometry (Fig. 3c, d and Supplementary Fig. 3a–c). Furthermore, FH-DHX36 specifically bound to the G4-forming consensus RRE in microscale thermophoresis experiments (Supplementary Fig. 3d).

In the following, we will focus our functional analysis on mRNA targets obtained by FH-DHX36-E335A PAR-CLIP, considering its high correlation with the FH-DHX36 PAR-CLIP (Fig. 2b), resulting in the identification of similar RREs (Fig. 3b), and its greater sequencing depth. Note that we obtained comparable results using the wild-type DHX36 PAR-CLIP data, as shown in the Supplementary Figures complementing the main functional analyses presented below.

Using this data, we asked whether DHX36 binding sites enriched at sequences in the human transcriptome that formed rG4s in vitro[10]. We found that 74% of the rG4s identified in the 3′ UTR were recovered in the FH-DHX36-E335A PAR-CLIP (Fig. 3e). 59% and 44% of the rG4 sites in 5′ UTR and CDS, respectively, also overlapped with FH-DHX36-E335A PAR-CLIP binding sites. Collectively, our in vivo and in vitro data showed that FH-DHX36 is preferentially binding thousands of mature mRNAs at G-rich elements in the CDS and UTRs, many of which were shown to form rG4s (Fig. 3f, Supplementary Fig. 3e), further supporting the hypothesis that DHX36 is acting on these structures in vivo.

**Loss of DHX36 activity leads to target mRNA stabilization.** RNA structures, including rG4s, impact RNA turnover, localization, and translation[5,16,17,34]. In order to investigate the gene regulatory roles of DHX36 in loss-of-function studies, we created DHX36-knockout (KO) HEK293 cells using Cas9 targeted to the DHX36 gene with a single guide RNA. Sequencing of the DHX36 genomic locus (Supplementary Fig. 4b) and western blotting of the clone used in follow-up experiments confirmed an extensive deletion and loss of detectable protein (Supplementary Fig. 4c). Although DHX36 overexpression showed little impact, DHX36 loss had a profound effect on the growth rate and morphology of HEK293 cells. KO cells proliferated at ~50% growth rate compared to parental HEK293 (Supplementary Fig. 4d) and cells appeared incapable of spreading evenly in the culture dish (Supplementary Fig. 4e), in agreement with a cell proliferation defect found in hematopoietic cells of conditional DHX36-KO mice[35]. This phenotype depended on the DHX36 helicase function and could be rescued by introduction of a FH-DHX36 transgene, but not by the mutant FH-DHX36-E335A, which even further reduced proliferation rates (Supplementary Fig. 4f, g).

Using DHX36-KO cells, we investigated the effect of DHX36 on target mRNA abundance using RNA-seq (Supplementary data 4). Loss of DHX36 led to an increase in target mRNA levels, dependent on the number of DHX36 binding sites (Supplementary Fig. 5a, c) or the number of crosslinked reads per target mRNA normalized by overall mRNA abundance (normalized crosslinked reads per million, NXPM) (Fig. 4a, Supplementary Fig. 5d). We previously found that both metrics correlated well with the occupancy of an RBP on its target[30,36,37]. For the top FH-DHX36-E335A targets binned by cluster number (>20 clusters, $n = 218$) or NXPM (NXPM > 100, $n = 381$) mRNA levels were increased upon DHX36 loss by 25 and 15%, respectively. By binning our targets according to DHX36 binding in the 3′ UTR, 5′ UTR, or CDS, we found that binding to the UTRs conferred a considerably stronger effect on mRNA abundance compared to CDS binding sites (Fig. 4b, c and Supplementary Fig. 5b, e, f, g). In the FH-DHX36-E335A PAR-CLIP we recovered additional AU-rich clusters in addition to the G-rich binding sites (Fig. 3a); however, in our analysis we were not able to tease out whether they also contributed to mRNA abundance changes, considering that we found no transcripts that showed robust AU-rich sites, without G-rich clusters nearby. Because of the large overlap of DHX36 binding sites with rG4s (Fig. 3e), we tested the effect of DHX36-KO on the abundance of rG4-mRNAs. Indeed, target mRNAs harboring an rG4s in vitro increased in abundance by ~16% in DHX36-KO cells, indicating that DHX36 was involved in their regulation (Fig. 4d). Finally, we confirmed with qPCR analysis that the levels of two different endogenous targets, WAC and PURB, increased by DHX36-KO (Fig. 4e and Supplementary Fig. 5h).

Next, we tested whether DHX36 required its helicase function for transcriptome remodeling and profiled mRNAs of DHX36-KO cells stably expressing FH-DHX36 or FH-DHX36-E335A (Supplementary Fig. 4f, Supplementary data 4). As expected, considering that DHX36-E335A was incapable of rescuing the cellular phenotype of DHX36 loss (Supplementary Fig. 4g), cumulative distribution analysis of DHX36 PAR-CLIP targets revealed that only expression of the wild-type construct was able to revert the effect of DHX36 loss on target mRNA abundance (Fig. 4f).

We reasoned that a direct posttranscriptional gene regulatory activity was the likeliest role for DHX36, considering its cytoplasmic localization and strong RNA-binding. We selected four DHX36 targets, WAC, PURB, NAA50, and SLMO2, that were among the top 100 DHX36 targets in our PAR-CLIP analysis and accumulated in DHX36-KO cells and found that

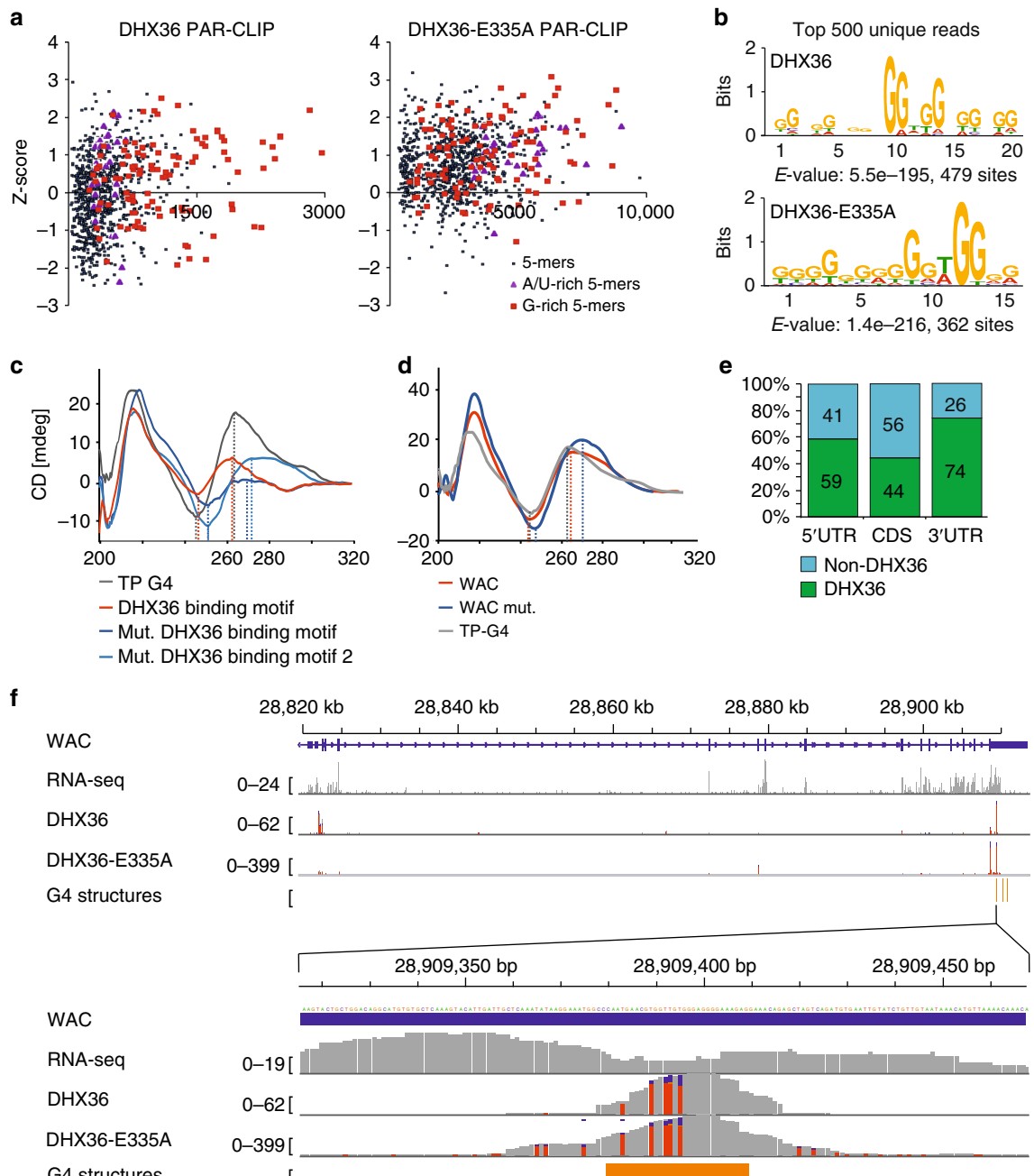

**Fig. 3** DHX36 recognizes quadruplex-forming G-rich sequence stretches on mRNA. **a** A comparison of Z-scores and occurrence of all possible 5mers shows an enrichment of G-rich sequences in FH-DHX36 PAR-CLIP binding sites (left panel). Same analysis for FH-DHX36-E335A PAR-CLIP (right panel) shows additional enrichment for A/U-rich 5mers. 5mers containing at least three Gs (red squares) or being A/U-rich (purple triangles) are highlighted. **b** Weblogo of the RNA recognition element of FH-DHX36 (top) and FH-DHX36-E335A (bottom) PAR-CLIP binding sites generated by MEME (*P*-value <0.0001) using the top 500 unique reads. **c** Circular dichroism spectra of oligonucleotides (Supplementary Table 1) after performing a G4-folding protocol. The FH-DHX36 PAR-CLIP-derived RRE (red) shifts towards peaks of positive control TP-G4 (gray), whereas two mutated binding motifs (light and dark blue) do not shift. Lines represent mean of ten subsequent measurements. **d** Same as **c** but with a native DHX36 RRE of the WAC mRNA (red) and a mutated version (blue) (Supplementary Table 1). Source data are provided as a Source Data file. **e** Percent of sites in the human transcriptome forming rG4s in vitro[10] categorized by 5′ UTR, CDS, and 3′ UTR found in DHX36 PAR-CLIP binding sites (green). **f** Top panel: screenshot of the FH-DHX36 and FH-DHX36-E335A PAR-CLIP binding sites for the representative target mRNA WAC. The gene structure is shown, as well as coverage from a HEK293 RNA-seq experiment. The bottom two tracks show the alignment of sequence reads with characteristic T-to-C mutations from a FH-DHX36 and FH-DHX36-E335A PAR-CLIP experiment. Bottom panel: close-up of the indicated region in the 3′ UTR of WAC. rG4s formed in vitro[10] are indicated in orange

their half-lives significantly increased in DHX36-KO cells (Fig. 4g, Supplementary Fig. 5i–k). Nevertheless, DHX36 was previously shown to unwind DNA and RNA G4s[25,27] and thus, we formally investigated whether the accumulation of target transcripts in DHX36-KO was partly due to an increase in their transcription.

Compared to wild-type cells DHX36-KO cells showed no increase in newly-synthesized target mRNAs (Fig. 4h and Supplementary data 6), profiled by sequencing of nascent chromatin-associated RNA[38]. Taken together, our results indicate that—at least in HEK293 cells—DHX36 regulates gene expression exclusively in a

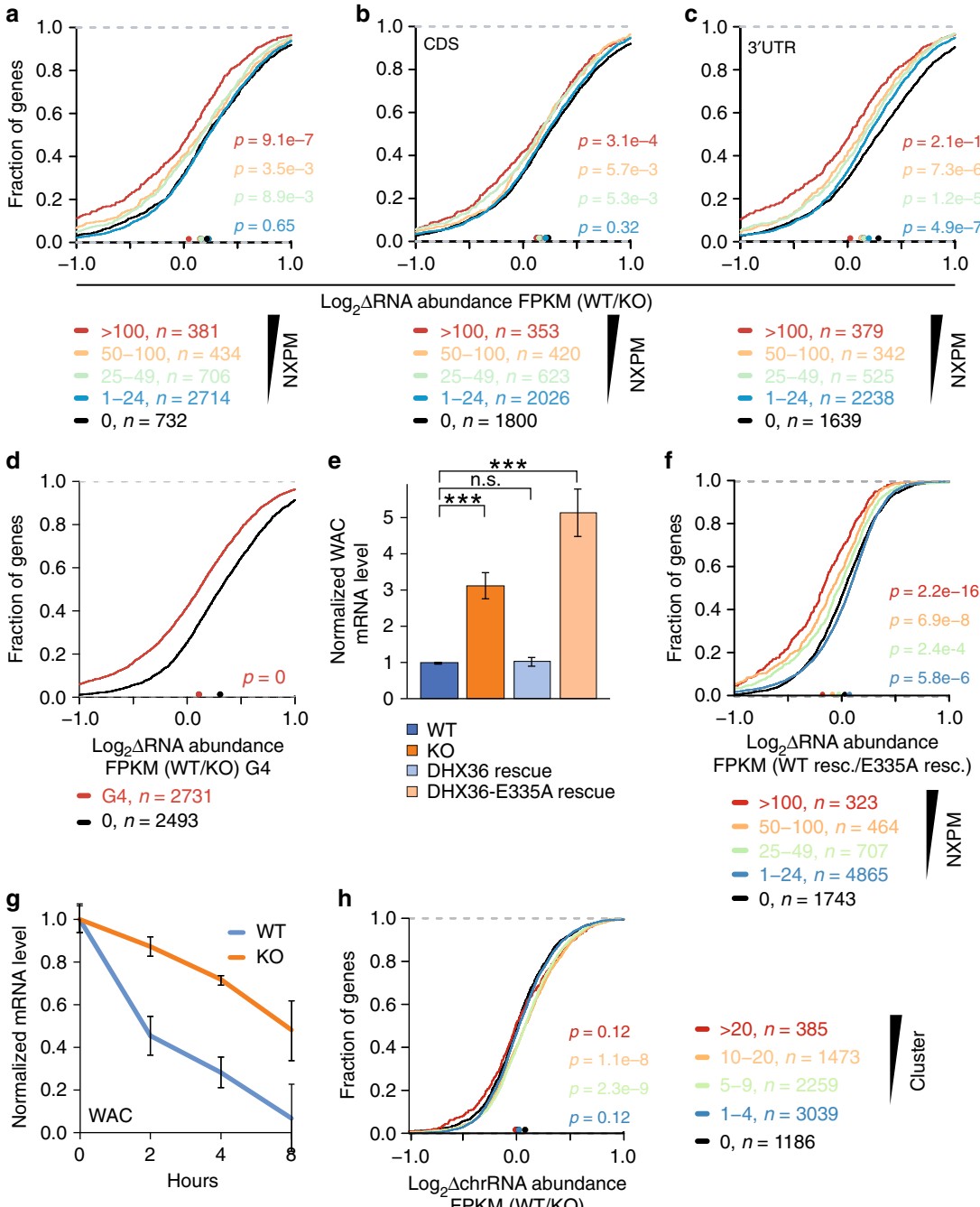

**Fig. 4** DHX36-KO results in increased target RNA abundance. **a** DHX36-KO results in an increased target mRNA abundance shown by CDF comparing changes in target mRNA abundance of DHX36-KO ($n = 3$) and parental HEK293 cells ($n = 3$). Target mRNAs were binned in accordance to the number of NXPM obtained by DHX36-E335A PAR-CLIP. Significance was determined using a two-sided Kolmogorov–Smirnov (KS) test. **b** Same as in **a**, except mRNAs were binned based on the number of NXPM in the CDS. **c** Same as in **a**, except mRNAs were binned based on the number of NXPM in the 3′ UTR. **d** Same as in **a**, except mRNAs were binned based on whether they harbor a G4-site identified previously[10] overlapping with PAR-CLIP binding sites. **e** Quantification of WAC mRNA levels in HEK293 wild-type cells (WT), DHX36-KO cells (KO), DHX36-KO cells with FH-DHX36 overexpression (DHX36 rescue), and DHX36-KO cells with FH-DHX36-E335A overexpression (DHX36-E335A rescue) by RT-qPCR. WAC mRNA levels were normalized to the level of U6 snRNA. Median WAC WT levels were scaled to 1. Significance was calculated using a Student's $t$-test ($n = 3$). Significance levels: $*P < 0.05$, $**P < 0.01$, and $***P < 0.001$ compared to normalized WT WAC level. Error bars represent standard deviations of three experiments. **f** Same as in **a**, except target mRNA abundance in HEK293 DHX36-KO cells with FH-DHX36 overexpression were plotted over target mRNA abundance in HEK293 DHX36-KO cells with FH-DHX36-E335A overexpression. **g** DHX36 target mRNA WAC show increased half-life upon DHX36-KO shown by qPCR after transcriptional block with actinomycin D and isolation of RNA at the indicated timepoints. Error bars represent standard deviations of three experiments. **h** DHX36 regulates target mRNA abundance at a posttranscriptional rather than transcriptional level shown by CDFs comparing changes in nascent target mRNA abundance purified from chromatin of DHX36 knockout cells ($n = 3$) and parental HEK293 cells ($n = 3$). Target mRNAs were binned in accordance to the number of binding sites obtained by DHX36-E335A PAR-CLIP. Significance was determined using a two-sided KS test. Source data are provided as a Source Data file

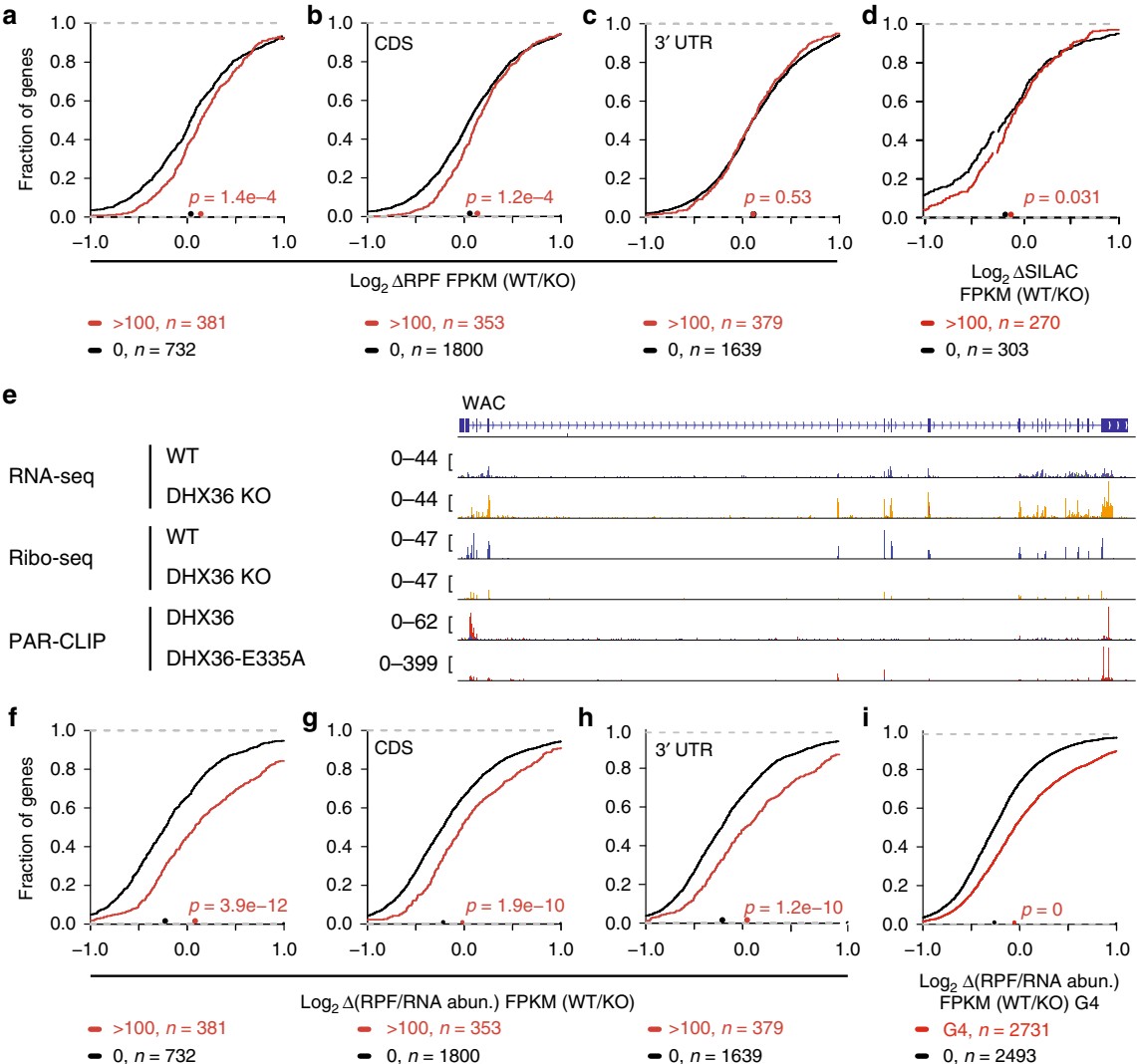

**Fig. 5** DHX36-KO results in reduced target mRNA translation efficiency. **a** Cumulative distribution function comparing changes in ribosome-protected fragments (RPFs) of DHX36-KO ($n = 3$) and parental HEK293 cells ($n = 3$). Target mRNAs are binned in accordance to the number of NXPM obtained by DHX36-E335A PAR-CLIP. Significance was determined using a two-sided KS test. **b** Same as in **a**, except target mRNAs were binned in accordance to NXPM in the CDS. **c** Same as in **a**, except mRNAs were binned based on the number of NXPM in the 3′ UTR. **d** Same as in **a**, except protein abundance changes as determined by SILAC were plotted. **e** Screenshot of RNA-seq and Ribo-seq coverage in wild-type and DHX36-KO HEK293 cells on the representative DHX36 target WAC. Bottom two tracks show the coverage for DHX36 and DHX36-E335A PAR-CLIP. **f** Cumulative distribution function comparing changes in translation efficiency (TE, RPF/RNA abundance) of DHX36-KO ($n = 3$) and parental HEK293 cells ($n = 3$). Target mRNAs are binned in accordance to the number of NXPM obtained by DHX36-E335A PAR-CLIP. Significance was determined using a two-sided KS test. **g** Same as in **f**, except mRNAs were binned based on the number of NXPM in the CDS. **h** Same as in **f**, except mRNAs were binned based on the number of NXPM in the 3′ UTR. **i** Same as in **f**, except mRNAs were binned based on whether they harbor a G4-site identified previously[10] overlapping with PAR-CLIP binding sites or not

posttranscriptional manner, and that its loss results in the stabilization of target mRNAs in a helicase-dependent manner.

**DHX36 increases translational efficiency of its targets**. Next, we asked whether the accumulation of target mRNA levels upon DHX36 loss resulted in a concomitant change in translation. We measured the impact of DHX36 on ribosome occupancy on its targets using ribosome footprinting[39] (Ribo-seq) in DHX36-KO and the corresponding parental cells (Supplementary data 5). Surprisingly, DHX36 loss resulted in a marginal, albeit statistically significant decrease in ribosome-protected fragments (RPFs) ($P < 10^{-5}$, two-sided Kolmogorov–Smirnov (KS) test, Fig. 5a–e and Supplementary Fig. 6a, c, d), particularly for sites in the CDS (Fig. 5b, c and Supplementary Fig. 6b, e, f). We also measured

changes in global protein levels using stable isotope labeling in cell culture (SILAC) followed by mass spectrometry. A modest but significant decrease in protein levels from DHX36 top targets upon DHX36 loss was detected, which is consistent with the decrease in RPFs (Fig. 5d and Supplementary Fig. 5h and Supplementary data 5).

We calculated the average density of ribosomes on each mRNA in DHX36-KO and control cells by normalizing the number of RPFs with the mRNA abundance. This score, known as the translational efficiency (TE), normalizes for mRNA abundance and approximates the translational output per mRNA molecule of a given gene[39]. DHX36-KO strongly correlated with a decreased TE on DHX36 targets (~27% decrease for the 381 top DHX36 targets with NXPM > 100, respectively) (Fig. 5f). Interestingly, the decrease in target TE upon DHX36-KO was again more

pronounced in targets bound in the UTRs than in the CDS (Fig. 5g, h and Supplementary Fig. 6i). We found that rG4-forming RNAs exhibited a 17% decreased TE upon DHX36-KO (Fig. 5i), further supporting a role of DHX36 in resolving rG4s.

We complemented our systems-wide data and studied the impact of DHX36 binding to selected target sites and generated reporter cells stably expressing an mCherry-coding transgene fused to G4-forming DHX36 PAR-CLIP binding sites and controls. We used the rG4-forming binding site in the WAC and PURB mRNA 3′ UTR and mutants that were not able to fold into an rG4 (Fig. 3d, Supplementary Fig. 3b), as well as two non target sequences from the DDX5 mRNA (Supplementary Fig. 7a). As expected, reporter protein expression did not change for the control plasmids, but was reduced upon DHX36-KO for the DHX36 wild-type cluster plasmids (Supplementary Fig. 7b). Mutation of the rG4-forming element in the DHX36 binding site made the reporters insensitive to DHX36-KO, strengthening our hypothesis that DHX36 promoted translation by unwinding of rG4 elements. Taken together with our observation that >90% of DHX36 proteins did not co-sediment with translating ribosomes (Fig. 1d) and thus unlikely influenced translation elongation, our data suggest that DHX36 increased the translational competence of mRNAs, possibly allowing access to the translational machinery either by resolving rG4s blocking translation initiation or by changing localization of target mRNAs.

**DHX36 unwinds rG4s to increase translational efficiency**. We hypothesized that if DHX36 functions in resolving rG4s and other structures on mRNA, the DHX36-KO will result in an increase in rG4 formation in living cells. Thus, we stained G4 structures in wild-type and DHX36-KO cells using the BG4 antibody specific for DNA and RNA G4[11,40] (Fig. 6a). We observed a strong BG4 signal from the nucleus that only marginally changed upon DHX36-KO and likely represented DNA G4 (Fig. 6a, b). In contrast, BG4 signal from the cytoplasm increased by ~1.5-fold, indicating an accumulation of rG4s upon DHX36-KO that corresponded in magnitude with the levels of DHX36 target mRNA stabilization (Fig. 6a–c). Whereas treating wild-type cells with carboxypyridostatin (cPDS), a small molecule that specifically stabilizes rG4s[11], resulted in even higher cytoplasmic BG4 signal compared to DHX36-KO, additional rG4s in DHX36-KO were sensitive to RNase A treatment (Fig. 6c, Supplementary Fig. 8), confirming that the cytoplasmic BG4 signal originated from RNA containing G4s.

Next, we tested whether the unresolved rG4s in target mRNAs upon DHX36 loss led to the observed mRNA stabilization. We profiled the transcriptome of HEK293 cells after treatment with cPDS and observed a significant increase in DHX36 target mRNA abundance, even exceeding the effect of DHX36 loss (Fig. 6d and Supplementary data 6). Taken together, our data suggest that DHX36 loss resulted in increased abundance of rG4s and that the formation of these structures stabilized the RNA without corresponding increase in translation.

**DHX36 loss activates the cellular stress response**. We hypothesized that reduced translation initiation upon DHX36 loss was unlikely to account for the observed reduced translational efficiency, considering that DHX36 binding in 3′ UTR was as efficient in promoting translation as binding in the 5′ UTR. More likely, rG4 and other structure formation in target mRNAs upon DHX36 loss resulted in their sequestration into translationally inactive subcellular compartments, such as stress granules (SG) or P-bodies. DHX36 itself was found to localize to SGs[22] and thus, we cross-referenced our PAR-CLIP data with a recently published dataset of transcripts enriched in SGs[41]. Indeed, DHX36 targets

and transcripts harboring rG4s[10] were among the mRNAs significantly enriched in SGs (Fig. 7a, b). Furthermore, DHX36-KO cells did show signs of cellular stress, such as reduced proliferation and morphological changes (Supplementary Fig. 3). This is in agreement with in vivo data suggesting an essential function of DHX36 in development[29,35,42]. Western blot analysis further revealed that the abundance of one of the markers of the cellular stress response, phosphorylated PKR/ElF2AK2 (phospho-PKR)[43], was significantly increased in DHX36-KO cells, even without stress induction (Fig. 7c), whereas levels of unphosphorylated PKR stayed normal (Supplementary Fig. 9). Phospho-PKR levels could be rescued by transgenic expression of FH-DHX36 but further increased by expression of the mutant FH-DHX36-E335A (Fig. 7c). In addition, DHX36-KO cells showed an higher percentage of stress granules in the absence of any stress stimuli, determined by fluorescence microscopy. This effect could be almost completely rescued by reintroduction the transgenic wild-type helicase into the DHX36-KO cells (Fig. 7d). Our data suggest that the increased rG4 formation resulting from DHX36-KO on one hand triggers the cellular stress response, and on the other hand leads to translational silencing of rG4-containing transcripts, possibly by sequestration in SGs and their seeds/precursors[44].

## Discussion

Here we present a comprehensive and systems-wide characterization of the targets and function of the DEAH-box helicase DHX36. We identified RNA-binding sites transcriptome-wide, delineated consensus binding motifs, and globally defined the effect of DHX36 loss on target mRNA abundance and translation. Before our systems-wide study a handful of DHX36 targets, which we largely recovered, were identified[24,26,29,45] (Supplementary data 3), however, the regulatory impact of DHX36 remained unclear. For example, it remained unresolved what cellular compartment DHX36 preferentially localizes to[24], whether it preferred DNA[25,27] or RNA[24] as ligands, and whether it specifically recognized G4 structures[23,46], or also AU-rich elements[24], leading to varying hypotheses about its cellular function. Our data clearly demonstrate that, at least in HEK293 cells, DHX36 is a cytoplasmic mRNA binding protein (Figs. 1 and 2). Although there may conceivably be cell-type-specific differences in protein localization, our data are in agreement with two recent mRNA interactome studies in HeLa and HEK293 cells, where DHX36 scored as an RBP binding polyadenylated RNA[47,48].

We were able to demonstrate that DHX36 preferentially bound G-rich binding motifs, that significantly overlapped with sites forming rG4 structures in vitro[10] (Fig. 3). Nevertheless, a DHX36 mutant with an inactive helicase (DHX36-E335A) domain also crosslinked at AU-rich sequences, suggesting that these sites may serve as additional recruitment platforms, but that the active protein rapidly translocates to the more structured G-rich regions being unwound. This is in line without observation that the vast majority of additional AU-rich binding sites in the DHX36-E335A are on mRNAs with additional G-rich binding motifs. Although ~2,000 of our sites were shown to form rG4s in vitro[10] and also fulfill predictive criteria for G4 folding, recent studies suggest that rG4s can form with less guanines and longer loops than previously estimated[18,33], possibly resulting in an underestimation of rG4s in our data.

In contrast to prokaryotes, in eukaryotes thousands of sites in the transcriptome form stable rG4s in vitro[9,10], but appear to be globally unfolded in vivo[9], leading to the proposal of a specialized machinery regulating their formation. Guo and Bartel were not able to detect rG4 formation by DMS-seq after DHX36 knockdown or partial ATP depletion and speculated that helicase-

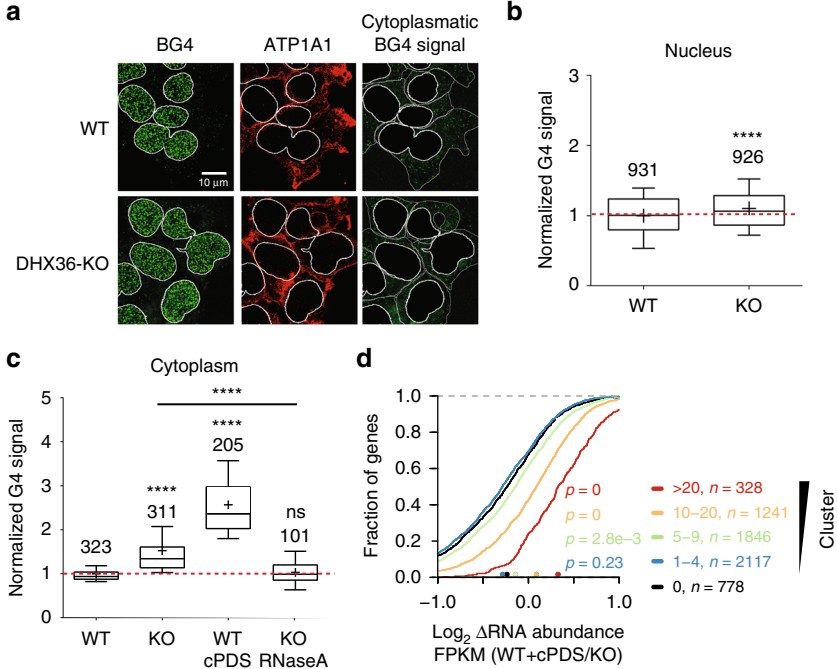

**Fig. 6** DHX36 is involved in resolving of rG4s in living cells. **a** DHX36 loss results in increased cytoplasmic rG4 signal, revealed by labeling of wild-type and DHX36-KO HEK293 cells with the BG4 antibody (green). Anti-alpha 1 sodium potassium ATPase (ATP1A1) served to mark the cytoplasm (red) and allow removal of the nuclear signal. Panels with masked nuclear signal allow visualization of the increased BG4 signal in the cytoplasm in DHX36-KO cells. Scale bar, 10 μm. **b** Nuclear G4 levels determined by fluorescence intensity of cells normalized over WT. Mean of two biological replicates. Numbers indicate analyzed nuclei. Horizontal lines and plus signs represent median and mean values, respectively. Error bars show the distribution from the 10th to 90th percentile. Significance was determined using a two-sided KS test (*$P < 0.05$; **$P < 0.01$; ***$P < 0.001$; ****$P < 0.0001$). **c** Same analysis as in **b**, except fluorescence intensity of the cytoplasmic compartment was calculated by subtracting the nuclear signal from total cell fluorescence. If applied, additional treatment of samples is indicated. **d** Stabilization of G4 structures using carboxypyridostatin (cPDS) in HEK293 cells results in the accumulation of DHX36-E335A PAR-CLIP targets to a larger degree than DHX36-KO as shown by CDFs comparing changes in target mRNA abundance of DHX36 knockout cells ($n = 3$) and parental HEK293 cells treated with cPDS ($n = 3$). Target mRNAs were binned in accordance to the number of binding sites obtained by DHX36-E335A PAR-CLIP. Significance was determined using a two-sided KS test. Source data are provided as a Source Data file

independent RBPs globally resolved rG4s[9]. Nevertheless, considering that at least eight different DEAH or DEAD-box helicases are candidates to interact with rG4s[19,49,50] and likely compensate for each other's loss, it remains unclear whether DMS-seq would be sensitive enough to reliably detect rG4 formation after the knockdown of a single factor. Compensation by other helicases is well-documented for DNA G4 in yeast, where Rrm3 is able to rescue Pif1 loss[51]. The fact that DHX36-KO cells remain viable with a growth defect and only exhibit an ~1.5-fold increase in rG4 detectable by immunofluorescence (Fig. 6) and an ~30% accumulation of the best DHX36 target mRNAs (Fig. 4) does hint at a larger network of rG4-resolving factors in vivo. The factors partially compensating for DHX36 loss remain to be identified; the two other rG4-unfolding factors characterized in a systems-wide manner, EIF4A1 and CNBP[5,20], can be excluded considering that they act on 5′ UTR and CDS rather than 3′ UTR and directly influence translation in contrast to DHX36. Finally, DHX36 prefers to unwind parallel G4 structures, exactly the kind typically formed on RNA, further supporting its posttranscriptional regulatory role in the cytoplasm[25].

Most studies focused on rG4 in mRNA 5′ UTR and CDS[5,16,17,20,52], and their possible impact on translation elongation[53], ribosomal frameshift[54], and no-go decay[55]. rG4 in 3′ UTR are studied less intensively, but have been implicated most prominently in cleavage and polyadenylation site selection, resulting in differential isoform expression[15]. To our knowledge our study provides the first link between regulation of rG4 formation in 3′ UTRs and mRNA stabilization (Fig. 4) and a simultaneous marked reduction in translational efficiency (Fig. 5). We excluded

a direct DHX36 effect on translation, considering that only a minority of DHX36 was found on polysomes and coverage of RPFs did not change in DHX36-KO HEK293 cells. This contrasts with a recent report that showed DHX36 associating with polysomes and affecting translation of small open reading frames in the 5′ UTR of HeLa cells, observations we did not see in our cells using thoroughly validated reagents, possibly reflecting the use of a different cell system[56]. Loss of DHX36 did not affect the distribution of ribosome density at either translation initiation, termination, or rG4 forming sites within the CDS, further indicating that DHX36 does not directly affect the translation machinery. Thus, the stabilized G-rich target mRNA in DHX36-KO cells were not translational competent, either due to decreased translation initiation, or by sequestration of these RNAs into granules, such as P-bodies or SGs (Fig. 8). Our data are more congruent with DHX36 target mRNAs accumulating in SGs or their precursors/seeds[44] that are difficult to detect by microscopy, considering (1) that these granules store untranslated mRNAs and may recruit rG4[22,57–59], (2) DHX36 itself is recruited to SGs[22], and (3) DHX36 target mRNAs in general and those that form rG4s in vitro in particular enrich in SGs (Fig. 7). Furthermore, DHX36-KO cells exhibited increased levels of SGs and of the cell stress marker phospho-PKR, implying that DHX36-KO cells were inherently stressed, while accumulating DHX36 target mRNAs and rG4s (Figs. 6 and 7). We speculate that unresolved rG4s themselves caused the stress response. This observation may also be linked to the role of a number of DExD/H-box helicases as RNA sensors[60] in the innate immune response triggering stress, or as essential host factors for the replication of viruses[60] that

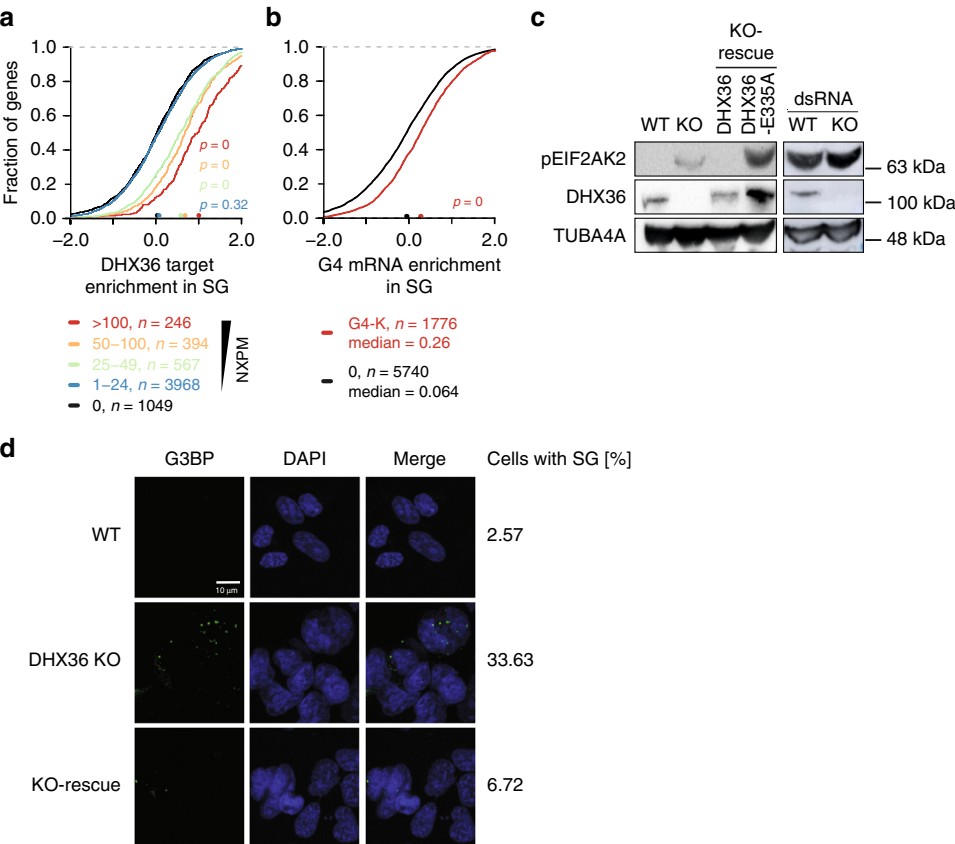

**Fig. 7** DHX36 mRNA targets are enriched in stress granules. **a** Cumulative distribution function showing enrichment of DHX36 mRNA target levels in stress granules compared to non-targets. Target mRNAs are binned in accordance to the number of NXPM obtained by DHX36-E335A PAR-CLIP. Significance was determined using a two-sided KS test. **b** Cumulative distribution function showing enrichment of G4-forming DHX36 mRNA target levels in stress granules compared to non-targets. Target mRNAs are binned based on whether they harbor a G4-site identified previously[10] overlapping with PAR-CLIP binding sites or not. Significance was determined using a two-sided KS test. **c** Western blot analysis of wild-type HEK293 cells (WT), DHX36-KO cells (KO), DHX36-KO cells with transgenic FH-DHX36 expression (DHX36-KO rescue), and DHX36-KO cells with FH-DHX36-E335A expression (DHX36-E335A-KO rescue). Positive control for PKR phosphorylation by dsRNA transfection is shown on the right. Source data are provided as a Source Data file. **d** DHX36 loss results in increased SG formation, revealed by labeling of wild-type, DHX36-KO, and DHX36-KO wild-type rescue HEK293 cells with a G3BP antibody (green). Stress granules were counted and divided by the number of cells. Scale bar, 10 μm

frequently contain G4s in their genomes[61]. In summary, our data serve as a comprehensive resource for studying target interactions of RNA helicases that are underrepresented in systems-wide interaction studies and provide a new link between rG4 formation and transcript stability on one hand and the stress response on the other hand.

## Methods

**Cell lines and cell culture**. Wild-type HEK293 T-Rex Flp-In cells (Thermo Fischer Scientific) were grown in Dulbecco's modified Eagle's medium (DMEM) (Thermo Fisher Scientific) supplemented with 10% (v/v) fetal bovine serum (FBS) (Thermo Fisher Scientific), 100 U ml$^{-1}$ Penicillin–Streptomycin (Thermo Fisher Scientific), 100 μg ml$^{-1}$ zeocin (InvivoGen), and 10 μg ml$^{-1}$ blasticidin (InvivoGen). Originating from these cells, transgenic cell lines with stable integration of constructs were generated by co-transfection of cloned FRT-plasmids (this study, see section "Plasmids") with the Flp recombinase expression vector pOG44 (Thermo Fisher Scientific) as previously described[62]. Positive clones of these cell lines were selected and cultured in the same media as described above except using 100 μg ml$^{-1}$ hygromycin (InvivoGen) instead of zeocin.

**Plasmids**. Full-length complementary DNA of wild-type HEK293 T-Rex Flp-In cells was used as template for cloning DHX36 into the pFRT/TO/FLAG/HA-Dest plasmid[62] using the restriction enzymes BamHI (Thermo Fisher Scientific) and XhoI (Thermo Fisher Scientific). The so-generated pFRT-FlagHA-DHX36-iso1 was used to create the plasmids pFRT-FlagHA-DHX36-iso2, pFRT-FlagHA-DHX36-iso1-E335A, and pFRT-FlagHA-DHX36-iso2-E335A by site-directed mutagenesis. For this, primer with the desired mutation were designed and set in a PCR with the paternal plasmid and 2× Phusion PCR Master Mix (Thermo Fisher Scientific).

Reporter gene plasmids were generated by cloning DHX36 binding sites of the WAC and PURB mRNA 3′ UTRs as well as two non-targeted regions of the DDX5 mRNA 3′ UTR into the pcDNA5-FRT-GFP-mCherry-3pGW backbone[63] (Addgene) using the commercial BP and LR clonase systems according to the manufacturer's instructions (Thermo Fisher Scientific). Mutated version were generated by site-directed mutagenesis.

**CRISPR/Cas9 gene editing for DHX36 knockout cells**. crRNAs were designed using https://benchling.com. Alt-R crRNA was ordered from IDT. 100 pmol Alt-R crRNA and 100 pmol Alt-R tracrRNA-ATTO 550 were denatured at 95 °C for 5 min and incubated at RT for 15 min to anneal both strands in a total volume of 100 μl in Nuclease-Free Duplex Buffer (IDT). 15 pmol annealed RNA were combined with 15 pmol Cas9 (IDT) and 5 μl Cas9 + reagent (Thermo Fisher Scientific) in Opti-MEM (Thermo Fisher Scientific) in a total volume of 150 μl and mixed well. In a second tube 125 μl Opti-MEM was combined with 7.5 μl CRISPRMAX (Thermo Fisher Scientific) and mixed well. After incubation at RT for 5 min the content of the two tubes were combined, mixed well, and transferred to a 6-well compartment containing wild-type HEK293 T-Rex Flp-In cells. The cells were seeded the previous day at a density of $4 \times 10^5$ cells ml$^{-1}$. After 48 h ATTO 550 positive cells were FACS-sorted and seeded at the density of up to 1 cell per well in a 96-well plate using standard medium described above. Single clones were expanded and analyzed for loss of DHX36 protein by western blot using an anti-DHX36 antibody (Santa Cruz Biotechnology).

**Western blot analysis**. For standard protein analysis protein lysates were separated on SDS-PAGEs and blotted on a Protan BA83 Nitrocellulose membrane (GE Healthcare). After saturating free binding sites with 5% milk powder in 1× TBS-T membrane was incubated with suitable primary antibodies overnight at 4 °C under constant agitation. After three times 5 min washing with TBS-T, membrane was

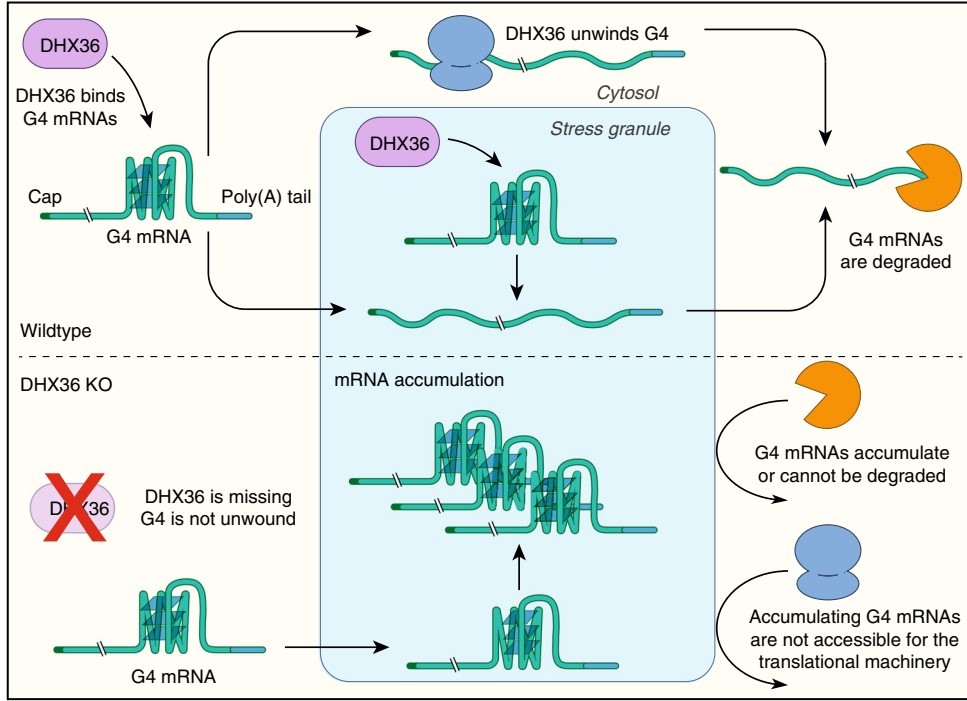

**Fig. 8** Schematic of our model of DHX36 function. DHX36 helps unwind G4 structures forming in the 3'UTR of mRNA and thus prevents their accumulation in SG and/or helps releasing them from these structures. In DHX36-KO cells, G4 containing mRNA cannot be translated and accumulate in SG

incubated with matching HRP-coupled secondary antibodies (anti-mouse or anti-rabbit (Santa Cruze Biotechnology)) for 1 h at RT followed by another three washing steps. Signals were detected by chemiluminescence of HRP-coupled anti-mouse or anti-rabbit secondary antibodies (Santa Cruz Biotechnology) on a Gel Doc XR + Gel Documentation System (Biorad). Uncropped blots are provided in the Source Data file.

Used antibodies: Anti-CANX (1:1000, Abcam ref# ab31290), Anti-DHX36 (1:500, Santa Cruz ref# sc-377485), Anti-FLAG (1:2000, Sigma-Aldrich ref# F1804), Anti-FMR1 (1:2000, Linder et al. 2008), Anti-HA (1:2000, Covance ref# MMS-101R), Anti-HISTH2B (1:2000, Abcam ref# ab1790), Anti-PKR/EIF2AK2 (1:1000, Abcam ref# ab32052), Anti-PKR/EIF2AK2 (phospho T446) (1:1000, Abcam ref# ab32036), Anti-RPL22 (1:2000, Santa Cruz ref# sc-136413), Anti-TUBA4A (1:5000, Sigma-Aldrich ref# T5168), HRP-conjugated goat anti-mouse (1:5000, Santa Cruz ref# sc-2031), and HRP-conjugated goat anti-rabbit (1:5000, Santa Cruz ref# sc-2030)

**Subcellular fractionation.** Transgene expression in FlagHA-DHX36-Iso1 and -Iso2 HEK293 cells was induced by addition of 500 ng ml$^{-1}$ tetracycline (Merck) for 15 h. After washing with ice-cold PBS (Thermo Fisher Scientific), induced cells were scraped off the 150-mm cell culture dish and collected by centrifugation. Unless otherwise stated, cell fractionation was performed as previously described[64]. In detail, pelleted cells were resuspended in 1 ml of hypotonic lysis buffer (HLB) (10 mM Tris, pH 7.5, 10 mM NaCl, 3 mM MgCl$_2$, 0.3% (v/v) NP-40, 10% (v/v) glycerol) per 75 mg cell pellet. After 10 min incubation on ice, cell lysate was briefly vortexed followed by 8 min centrifugation at 800 g and 4 °C. The cytoplasmic fraction (supernatant) was thoroughly transferred to a new tube and 5 M NaCl was added to a final concentration of 150 mM. The remaining nuclear fraction (pellet) was carefully washed four times with HLB. After washing, the pellet was resuspended in nuclear lysis buffer (NLB) (20 mM Tris, pH 7.5, 150 mM KCl, 3 mM MgCl$_2$, 0.3% (v/v) NP-40, 10% (v/v) glycerol) and sonicated in two cycles (40% power, 30 s ON, 2 min OFF, Branson sonifier W250-D). Both the cytoplasmic and the nuclear fraction were 15 min centrifuged at 18,000×g and 4 °C to remove all debris. Obtained supernatants were subject of further investigation by standard western blotting. Used markers for subcellular compartments: nuclear = anti-Histone 2B antibody (Abcam), cytoplasm = anti-α-Tubulin antibody (Merck), endoplasmic reticulum membrane = anti-Calnexin antibody (Abcam).

**Oligo-d(T) pulldown.** Wild-type HEK293 T-Rex Flp-In cells were grown on two 150-mm cell culture dishes washed with ice-cold PBS, and crosslinked by irradiation with 0.15 J cm$^{-2}$ 254 nm UV-light. Cells were scraped off the dishes and collected by centrifugation. Cell pellets were resuspended in 1.5 ml LiDS lysis buffer (20 mM Tris-HCl pH 7.5, 500 mM NaCl, 0.5% LiDS, 1 mM EDTA, pH 7.5, 5 mM DTT) and passed 3x through a 26-G-needle for shearing. After 10 min incubation

on ice, input samples were taken and in lysis buffer equilibrated oligo-d(T) magnetic beads (New England Biolabs) were added to the lysate. Binding of poly-adenylated RNAs to the oligo-d(T) beads was performed for 1 h at 4 °C under constant agitation. Beads were collected on a magnetic rack and washed twice with wash buffer 1 (20 mM Tris-HCl pH 7.5, 500 mM NaCl, 0.1% LiDS, 1 mM EDTA pH 7.5, 5 mM DTT), wash buffer 2 (20 mM Tris-HCl pH 7.5, 500 mM NaCl, 1 mM EDTA, pH 7.5), and wash buffer 3 (20 mM Tris-HCl pH 7.5, 200 mM NaCl, 1 mM EDTA pH 7.5), respectively. Elution was achieved by incubation with 100 µl elution buffer (20 mM Tris-HCl pH 7.5, 1 mM EDTA, pH 7.5) for 3 min at 55 °C. Eluate was concentrated using a Speedvac Concentrator (Eppendorf) and mRNA binding of proteins was analyzed by standard western blotting.

**Polysome profiling.** Wild-type HEK293 T-Rex Flp-In cells were grown on a 150-mm cell culture dish to 90–100% confluency. Growth media was changed to media containing 25 µg ml$^{-1}$ cycloheximide (Merck). After 10 min incubation, cells were washed once with ice-cold PBS and 100 µl of polysome lysis buffer (20 mM Tris, pH 7.5, 100 mM KCl, 5 mM MgCl$_2$, 1 mM DTT, 0.5% (v/v) NP-40, 100 µg ml$^{-1}$ cycloheximide, 20 U ml$^{-1}$ SUPERaseIn, protease inhibitors) were added (note: for samples used for RNase-treated lysates, no SUPERaseIn was added). Cells were scraped of the dish and transferred to a pre-chilled 1.5 microcentrifuge tube. After 10 min incubation on ice, lysate was cleared by 10 min centrifugation at 20,000×g and 4 °C. Clarified lysate was loaded onto a 5–45% linear sucrose gradient (sucrose in 20 mM Tris, pH 7.5, 100 mM KCl, 5 mM MgCl$_2$) and centrifuged for 60 min in a SW60Ti rotor (Beckman) at 150,000×g and 4 °C. During fractionation using a Gradient fractionator (Biocomp) the UV profile (254 nm) was measured. Obtained fractions were further analyzed by standard western blotting.

**PAR-CLIP.** Photoactivatable-ribonucleoside-enhanced crosslinking and immuno-precipitation (PAR-CLIP) was performed with minor modifications as described previously[30]. Essential steps are described in the following. HEK293 T-Rex Flp-In DHX36 and DHX36-E335A cells were grown on 15- to 150-mm-cell culture dishes to 80% confluency. Induction of transgene expression (addition of 500 ng ml$^{-1}$ tetracycline (Merck)) was performed for 15 h together with feeding the cells with 100 µM of 4-thiouridin (4SU). After washing with ice-cold PBS cells were cross-linked (irradiation with 365 nm UV-light, 5 min) and scraped off the dishes using a rubber policeman. After pelleting by centrifugation cells were resuspended in 7 ml NP-40 lysis buffer (50 mM HEPES, pH 7.5, 150 mM KCl, 2 mM EDTA, 0.5 mM DTT, 0.5% (v/v) NP-40, protease inhibitors) and incubated on ice for 12 min. Cell lysate was clarified by 15 min centrifugation at 20,000×g and 4 °C. First RNase T1 (Thermo Fisher Scientific) digestion (1 U µl$^{-1}$) was performed for 15 min at 22 °C. 75 µl ml$^{-1}$ FLAG-M2 antibody (Merck) conjugated magnetic DynabeadsProtein G (Thermo Fisher Scientific) were added. Antigen capture was performed for 105 min at 4 °C on a rotating wheel. Beads were collected on a magnetic rack and washed 3×

with NP-40 lysis buffer. For trimming of the co-captured RNA, a second RNase T1 digestion (10 U µl$^{-1}$) was performed for 15 min at 22 °C with occasional shaking. 3′ ends of the RNA fragments were dephosphorylated using 0.5 U µl$^{-1}$ calf intestinal phosphatase (New England Biolabs) for 10 min at 37 °C, shaking. RNA was radioactively 5′-end-labeled using 1 U µl$^{-1}$ T4 polynucleotide kinase (Thermo Fisher Scientific) and 0.5 µCi µl$^{-1}$ $^{32}$P-γ-ATP (Hartmann Analytik) for 30 min at 37 °C. Almost complete 5′-end phosphorylation of RNA was accomplished by adding ATP (Thermo Fisher Scientific) to a concentration of 100 µM for 5 min. So-treated RNA-protein complexes were separated by SDS-Polyacrylamide Gel Electrophoresis (SDS-PAGE, NuPAGE 4–12% Bis-Tris Protein Gels (Thermo Fisher Scientific)). For detection, a blanked storage phosphor screen (GE Healthcare) was exposed to the gel and RNA-protein complexes of expected size were excised. After gel elution, protein components of the complexes were digested with 5 mg ml$^{-1}$ proteinase K (Carl Roth) for 2 h at 55 °C. Resulting RNA fragments were isolated by phenol–chloroform–isoamyl alcohol (25:24:1) extraction and subject of a small RNA cDNA library preparation protocol as previously described[30]. Here, a pre-adenylated, barcoded 3′-adapter oligonucleotide (rApp-TAA-TATCGTATGCCGTCTTCTGCTTG) was used.

So-obtained PAR-CLIP cDNA libraries were sequenced on an Illumina HiSeq 2500 platform (Illumina). Clusters of overlapping sequence reads mapped against the human genome version hg19 were generated using the PARalyzer software[31] incorporated into a pipeline (PARpipe, https://ohlerlab.mdc-berlin.de/software/PARpipe_119/) with default settings. Binding sites were categorized using the Gencode GRCh37.p13 GTF annotation (gencode.v19.chr_patch_hapl_scaff.annotation.gtf, http://www.gencodegenes.org/releases/19.html).

**5mer Z-scoring.** The BEDTools utility, *getfasta*, was used to recover the genomic nucleotide sequences corresponding to the remaining intervals in these BED files. To produce background, we used an in-house script to scramble the target read sequences while preserving GC content. We then counted the 5mers for target and for background, respectively, and calculated Z-score enrichment by proportion, $Z = \frac{X - \mu}{\sigma}$. X is the proportion $\frac{Count}{Total\ counts}$ for a given 5mer. Respectively, $\mu$ and $\sigma$ are the average and standard deviation of 5mer proportions in background.

**Analysis of the RNA recognition element.** Motif analysis was carried out using the MEME suite[65] using the top 500 unique cluster sequences defined by PARalyzer[31].

**Circular dichroism (CD).** For G-quadruplex formation, 10 µM oligodeoxynucleotides (Merck), resembling the DHX36 RREs, in folding buffer (10 mM Tris, pH 7.5, 100 mM KCl, 1 mM EDTA) were incubated in folding buffer (10 mM Tris, pH 7.5, 100 mM KCl, 1 mM EDTA) for 10 min at 95 °C, followed by subsequent passive cooling to RT. CD measurements were performed using a Jasco J-810 spectropolarimeter in 0.2 ml quartz cuvettes in the range of 200–350 nm. Measurements were averaged between ten accumulations with an instrument scanning speed of 10 nm s$^{-1}$.

**Microscale thermophoresis (MST).** For microscale thermophoresis, binding reactions were prepared in 1× MST buffer, supplemented with 0.5% BSA and 5 mM DTT in a total volume of 40 µl. A constant concentration of 25 nM 5′Cy5 labeled oligonucleotides (Merck) (folded) were used. A 1:1 serial dilution of DHX36, with 50 nM as highest concentration was used. In standard treated capillaries (NanoTemper Technologies) microscale thermophoresis analysis was performed with 80% LED, 20% MST power, on the Monolith NT.115 (Nano-Temper Technologies). Figures were made with Hill fit using the MO Affinity analysis software.

**Light microscopy.** Wild-type HEK293 T-Rex Flp-In cells and HEK293 T-Rex Flp-In DHX36 KO cells were seeded in 60 mm cell culture plates ($1.2 \times 10^6$). Microscopic images were taken 24, 48, 72, and 96 h after seeding using an EVOS FL cell imaging system (Thermo Fisher Scientific).

**Growth curves.** Wild-type HEK293 T-Rex Flp-In cells and HEK293 T-Rex Flp-In DHX36 KO cells were seeded in 35-mm cell culture dishes ($0.1 \times 10^6$). At indicated timepoints, cells were counted in triplicates using a Fuchs-Rosenthal-hemocytometer.

**Cell proliferation assay.** Proliferation assays were conducted using the commercial CellTiter 96® Non-Radioactive Cell Proliferation Assay (MTT) (Promega) according to the manufacturer's instructions. Short, wild-type HEK293 T-Rex Flp-In cells, HEK293 T-Rex Flp-In DHX36 -O cells, HEK293 T-Rex Flp-In DHX36-KO DHX36 rescue cells, and HEK293 T-Rex Flp-In DHX36 KO DHX36-E335A rescue cells were seeded to $5 \times 10^4$ in 96-well cell culture plates. Transgene expression was induced by addition of 500 ng ml$^{-1}$ tetracycline (Merck). 24, 48, and 72 h after seeding, cells were treated with the tetrazolium salt dye solution. 4 h later, solubilization solution was added and plates were kept in the dark at room temperature.

Absorbance at 570 nm was measured against a reference line at 700 nm on a Tecan infinite®200 plate reader.

**Flow cytometry.** For mCherry expression, cell lines with stably integrated pcDNA5-FRT-GFP-mCherry-3pGW reporter constructs were induced with 500 ng ml$^{-1}$ tetracycline (Merck) for 15 h. Cells were trpysinized, washed twice with PBS, and filtered through a 35 µm nylon net. Detection and analysis of the fluorescent signals of mCherry were performed on a LSRFortessa system (BD Biosciences). Triplicates of 100,000 double-positive cells were analyzed.

**RNA preparation for sequencing.** Transgene expression in HEK293 T-Rex Flp-In DHX36-KO DHX36 rescue cells and HEK293 T-Rex Flp-In DHX36-KO DHX36-E335A rescue cells was induced by addition of 500 ng ml$^{-1}$ tetracycline (Merck). In triplicates, RNA from the induced before-mentioned cell lines, wild-type HEK293 T-Rex Flp-In cells, and HEK293 T-Rex Flp-In DHX36-KO cells was isolated with TRIzol (Thermo Fisher Scientific) according to the manufacturer's instructions. Depletion of ribosomal RNA was accomplished by using the NEBNext rRNA Depletion Kit (New England Biolabs).

**RNA-sequencing preparation and analysis.** Complementary DNA preparation of the rRNA-depleted RNA was done with the NEBnext Ultra Directional RNA Library Prep Kit for Illumina (New England Biolabs). cDNA enrichment was facilitated using indexed primers of the NEBNext Multiplex Oligos for Illumina (New England Biolabs) and sequencing was performed on a HiSeq 2500 platform (Illumina). Sequencing reads were aligned to the hg19 human genome using Tophat 2 (ref. [66]). Cufflinks[66] was used to quantify reads on the UCSC hg19 annotation set differential expression was determined by Cuffdiff[66].

**Standard quantitative PCR.** Total RNA from human cells was isolated using TRIzol (Thermo Fisher Scientific) according to the manufacturer's instructions. Isolated RNA was reverse-transcribed using the QuantiTect Reverse Transcription Kit (Qiagen). Quantification of mRNA levels was done by qPCR using gene specific primer on a CFX96 Real Time System with a C1000 Touch Thermal Cycler (Biorad) or an Applied Biosystems StepOnePlus Real-Time PCR System (Thermo Fisher Scientific). As a fluorescent dye SYBR Green in an iQ SYBR Green Supermix (Biorad) was used. For normalization primer specific for U6 RNA were used.

**mRNA half-life determination.** HEK293 T-Rex Flp-In wild-type cells and HEK293 T-Rex Flp-In DHX36-KO cells were grown on 35-mm dishes to 80% confluency. At timepoint $t = 0$, media was changed to media containing 3 µg ml$^{-1}$ Actinomycin D (Merck). Cells were collected at timepoints 0, 2, 4, and 8 h after addition of Actinomycin D. mRNA levels were analyzed by standard quantitative PCRs.

**Preparation of chromatin-associated RNA for sequencing.** In triplicates, HEK293 T-Rex Flp-In wild-type cells and HEK293 T-Rex Flp-In DHX36-KO cells were grown on 150-mm-cell culture dishes to 80–90% confluency. After collecting the cells by centrifugation. Obtained pellets were washed once with ice-cold PBS. Cells were resuspended in 400 µl ice-cold Cytoplasmic Fraction Buffer (20 mM HEPES, pH 7.6, 2 mM MgCl$_2$, 10% (v/v) glycerol, 0.5 mM DTT, 0.1% (v/v) NP-40, protease inhibitors, 40 U ml$^{-1}$ Murine RNase Inhibitor (New England Biolabs)) and incubated on ice for 5 min. Crude lysates were layered on a 400 µl sucrose buffer cushion (10 mM HEPES, pH 7.6, 10 mM NaCl, 1.5 mM MgCl$_2$, 10% (v/v) glycerol, 37% (w/v) sucrose, 0.5 mM EDTA; 0.5 mM DTT, protease inhibitors, 40 U ml$^{-1}$ Murine RNase Inhibitor) and centrifuged for 20 min at 20,000×g and 4 °C. Resulting supernatant, representing the cytoplasmic fraction was collected and the pellet was resuspended in 200 µl ice-cold Nuclear Lysis Buffer (10 mM HEPES, pH 7.6, 100 mM NaCl, 50% (v/v) glycerol, 0.5 mM EDTA; 0.5 mM DTT, protease inhibitors, 40 U ml$^{-1}$ Murine RNase Inhibitor). 200 µl ice-cold 2 × NUN-Buffer (50 mM HEPES, pH 7.6, 600 mM NaCl, 7.5 mM MgCl$_2$, 2 M urea, 0.2 mM EDTA; 0.5 mM DTT, 2% (v/v) NP-40) were added dropwise, followed by a pulsed vortex step and incubation on ice for 20 min. After 30 min centrifugation at 20,000 × g and 4 °C, resulting supernatant (nucleoplasmic fraction) was collected and the pellet (chromatin) was washed twice with1 ml nuclear lysis buffer. The chromatin was resuspended in 1 ml TRIzol (Thermo Fisher Scientific) and the obtained mixtures were heated to 65 °C and run through a 26-G-needle until no insoluble objects were visible. Chromatin-associated RNA was isolated according to the manufacturer's instructions for TRIzol-based RNA isolation. RNA-seq library preparation and sequencing analysis was performed as described above.

**Ribosome footprinting.** Ribosome footprinting was performed as published previously[39]. Briefly, triplicates for both HEK293 T-Rex Flp-In wild-type cells and HEK293 T-Rex Flp-In DHX36-KO cells were grown to 80% confluency on 100-mm-cell culture dishes. For inhibition of translation, standard growth media was changed to growth media containing 100 µg ml$^{-1}$ cycloheximide (Merck). After 30 s media was aspirated and dishes were immediately cooled down on ice and washed with ice-cold PBS. 400 µl ribosome footprinting buffer (20 mM Tris, pH 7.4, 150 mM NaCl, 4 mM MgCl$_2$, 1 mM DTT, 100 µg ml$^{-1}$ cycloheximide, 1% NP-40, 25 U

ml$^{-1}$ Turbo DNase I (Thermo Fisher Scientific)) were added, cells were scraped off the dish using a rubber policeman. After 10 min incubation on ice, lysates were triturated by passing 10× through a 26-G-needle and cleared by 10 min centrifugation at 20,000×$g$ and 4 °C. For digestion of unbound RNA, 300 µl of cell extract was treated with 2.5 U µl$^{-1}$ RNase I (Thermo Fisher Scientific) and incubated at room temperature for 45 min with gentle mixing. RNase digestion was stopped by adding 0.65 U µl$^{-1}$ SUPERaseIn (Thermo Fischer Scientific) and extract was centrifuged through a 900 µl sucrose cushion (20 mM Tris, pH 7.4, 150 mM NaCl, 4 mM MgCl$_2$, 1 mM DTT, 100 µg ml$^{-1}$ cycloheximide, 1 M sucrose, 20 U ml$^{-1}$ SUPERaseIn) for 4 h at 200,000×$g$ (TLA 100.3 rotor) and 4 °C. The supernatant was carefully aspirated, and ribosome-containing pellet was resuspended in 150 µl of ribosome footprinting buffer supplemented with 20 U ml$^{-1}$ SUPERaseIn. RNA footprints were purified by phenol–chloroform extraction and precipitated with ethanol. After washing twice with 75% ethanol, the air-dried RNA pellet was resolved in 15 µl DEPC-treated water. Small RNA cDNA libraries for next-generation sequencing were prepared as previously described[30] with following modifications: firstly, for dephosphorylation 500 ng RNA was incubated 30 min at 37 °C with 0.7 U µl$^{-1}$ calf intestinal phosphatase (New England Biolabs). Afterwards, samples were separated on a 15% urea-PAGE for selection of 20–35 ribonucleotide long fragments. Extracted, precipitated and washed RNA fragments were ligated with preadenylated, barcoded 3′ adapters. Next, RNA was precipitated, washed and phosphorylated at the 5′-end with 5 U µl$^{-1}$ T4 polynucleotide kinase (Thermo Fisher Scientific) in 1× T4 DNA ligase buffer (20 µl reaction volume) for 30 min at 37 °C. Samples were separated on a 15% urea-PAGE for selection of 3′-ligated fragments. Extracted, precipitated, and washed RNA was ligated with 5′ adapters and samples were separated on a 12% urea-PAGE for selection of 5′- and 3′-ligated fragments. After reverse transcription and cDNA library enrichment, samples were sequenced on an HiSeq 2500 platform (Illumina).

After sequencing the reads were aligned to the human genome version hg19 using TopHat[66] and quantified on RNA defined in the UCSC hg19 annotation database using Cufflinks[66]. Overlaps of DHX36 cluster and different genomic regions were calculated with BEDTools[67].

**SILAC**. In duplicates, HEK293 T-Rex Flp-In wild-type cells and HEK293 T-Rex Flp-In DHX36 KO cells were grown to ~60% confluency on 35 mm-cell culture dishes. Then, cells were fed with light media (DMEM, 10% dialyzed FBS, 4 mM glutamine, 1.74 mM L-proline, 0.8 mM L-lysine, 0.4 mM L-arginine). 24 h later, cells were washed twice with pre-warmed PBS and wild-type cells were changed to medium-heavy media (DMEM, 10% dialyzed FBS, 4 mM glutamine, 1.74 mM L-proline, 0.8 mM L-lysine [4,4,5,5-D4], 0.4 mM L-arginine [U-13C6]) and KO cells to heavy media (DMEM, 10% dialyzed FBS, 4 mM glutamine, 1.74 mM L-proline, 0.8 mM L-lysine [U-13C6, 15N2], 0.4 mM L-arginine [U-13C6, 15N4]). 24 h later, cells were washed twice with pre-warmed PBS and cells were collected in 100 µl NP-40 lysis buffer (50 mM HEPES, pH 7.5, 150 mM KCl, 2 mM EDTA, 0.5 mM DTT, 0.5% (v/v) NP-40, protease inhibitors). After incubation for 10 min on ice, lysates were cleared by 15 min centrifugation at 20,000 × $g$ and 4 °C. Protein concentration was assessed by standard Bradford assay.

For in-gel digestion proteins were reduced and alkylated prior to SDS-PAGE by heating the cell lysates for 10 min at 70 °C in NuPAGE LDS sample buffer (Thermo Fisher Scientific) supplemented with 50 mM DTT. Samples were alkylated by adding 120 mM iodoacetamide Simply Blue (Thermo Fisher Scientific). Whole lanes were cut into 15 bands. The bands were destained with 30% acetonitrile, shrunk with 100% acetonitrile, and dried in a vacuum concentrator. Digests with 0.1 µg trypsin (Promega) per gel band were performed overnight at 37 °C in 50 mM ammonium bicarbonate buffer. Peptides were extracted from the gel slices with 5% formic acid. NanoLC-MS/MS analyses were performed on an Orbitrap Fusion (Thermo Fisher Scientific) equipped with an EASY-Spray Ion Source or a PicoView Ion Source (New Objective) and coupled to an EASY-nLC 1000 (Thermo Fisher Scientific). Using the Easy-Spray Ion Source the peptides were loaded on a trapping column (2 cm × 75 µm ID, PepMap C18, 3 µm particles, 100 Å pore size) and separated on an EASY-Spray column (25 cm × 75 µm ID, PepMap C18, 2 µm particles, 100 Å pore size). Using the PicoView Ion Source the peptides were loaded on capillary columns (PicoFrit, 30 cm × 150 µm ID, New Objective) self-packed with ReproSil-Pur 120 C18-AQ, 1.9 µm (Dr. Maisch). A 60-minute linear gradient from 3 to 40% acetonitrile and 0.1% formic acid was used. Both MS and MS/MS scans were acquired in the Orbitrap analyzer with a resolution of 60,000 for MS scans and 15,000 for MS/MS scans. HCD fragmentation with 35% normalized collision energy was applied A Top Speed data-dependent MS/MS method with fixed cycle time of 3 sec was used. Dynamic exclusion was applied with a repeat count of 1 and exclusion duration of 120 s; singly charged precursors were excluded from selection. Minimum signal threshold for precursor selection was set to 50,000. Predictive AGC was used with an AGC target value of 5e5 for MS scans and 5e4 for MS/MS scans. EASY-IC was used for internal calibration.

For MS raw data file processing, database searches and quantification, MaxQuant[68] version 1.5.7.4 was used. UniProt human reference proteome database was used in combination with a database containing common contaminants as a reverse concatenated target-decoy database. Protein identification was under control of the false-discovery rate (<1% FDR on protein and peptide level). In addition to MaxQuant default settings, the search was performed with tryptic cleavage specificity with three allowed missed cleaves. The search was performed with the following variable modifications: Gln to pyro-Glu formation and oxidation (on Met). Normalized H/M ratios were used for protein quantitation (at least two peptides per protein).

**Detection of G4 with BG4 antibody**. BG4 antibody was expressed and purified as described previously[11,48]. Cells were seeded in 6-Multiwell plate coated with poly-D-lysine solution. 48 h post seeding cells were pre-fixed with a solution 50% DMEM and 50% methanol/acetic acid (3:1) at RT for 10 min. Treatment with 2 µM cPDS performed for 24 h. After a brief wash with methanol/acetic acid (3:1), cells were fixed with methanol/acetic acid (3:1) at RT for 10 min. Cells were then permeabilized with 0.1% Triton X-100 in PBS at RT for 3 min. For RNase treatment, coverslips were incubated with 100 µg ml$^{-1}$ RNase A for 1 h at 37 °C. Cells were exposed to blocking solution (2% milk in PBS, pH 7.4) for 1 h at RT and then incubated with 2 µg per slide of BG4 and Anti-alpha 1 Sodium Potassium ATPase (Abcam ref #7671) antibodies in blocking solution (2 h at RT). Cells were then incubated with 1:800 of a rabbit antibody against the DYKDDDDK epitope (Cell Signaling ref# 2368) in blocking solution for 1 h. Next, cells were incubated at RT with 1:1000 Alexa Fluor 488 goat anti-mouse IgG (Life technologies ref# A11001) and Cyanine 3 goat anti-rabbit IgG (Life technologies ref# A10520) in blocking solution for 1 h. After each step, cells were washed three times for 5 min with 0.1% Tween-20 in PBS under gentle rocking. The cover glasses were mounted with Fluoroshield mounting media (Merck ref# F6057) containing DAPI (for nuclear staining). Slides were visualized at room temperature (RT) by using a confocal microscope (Leica SP5 AOBS).

**Detection of stress granules**. Cells were seeded in 6-Multiwell plate coated with poly-D-lysine solution. 48 h post seeding cells were pre-fixed with a solution 50% DMEM and 50% methanol/acetic acid (3:1) at RT for 10 min. After a brief wash with methanol/acetic acid (3:1), cells were fixed with methanol/acetic acid (3:1) at RT for 10 min. Cells were then permeabilized with 0.1% Triton X-100 in PBS at RT for 3 min. Cells were exposed to blocking solution (2% milk in PBS, pH 7.4) for 1 h at RT and then incubated with 5 µg ml$^{-1}$ G3BP1 antibody (BD ref # 611127) in blocking solution (2 h at RT). Next, cells were incubated at RT with 1:1000 Alexa Fluor 488 goat anti-mouse IgG (Life technologies ref# A11001). After each antibody stain cells were washed three times for 5 min with 0.1% Tween-20 in PBS under gentle rocking. The cover glasses were mounted with Fluoroshield mounting media (Merck ref# F6057) containing DAPI (for nuclear staining). Slides were visualized at room temperature (RT) by using a confocal microscope (Leica SP5 AOBS).

**Cell image analysis**. Fluorescence signal was determined using ImageJ software with the following formula: Corrected Total Cell Fluorescence (CTCF) = Integrate density − (area of selected cell × mean fluorescence of background readings). The cytoplasmic compartment was calculated by subtracting the nuclear signal, obtained by DAPI staining, from the total cell signal (Anti-alpha 1 Sodium Potassium ATPase). For graphical representation of signal distribution, we used box-and-whisker plots using GraphPad Prism 6 software with the following settings: boxes: 25–75 percentile range; whiskers: 10–90 percentile range. Statistical significance was determined by "Kolmogorov–Smirnov" and "Mann–Whitney" non-parametric tests. Data analyses were performed with Excel and GraphPad, and all figures were prepared with Adobe Illustrator.

**Preparation of cPDS-treated RNA for sequencing**. HEK293 T-Rex Flp-In wild-type cells were grown to ~50% confluency on 35-mm cell culture dishes. Then in triplicates, growth media was changed to standard media or media containing 2 µM carboxypyridostatin (cPDS) (Merck). After 42 h incubation, cells were collected, and total RNA was isolated using TRIzol according to the manufacturer's instructions. Depletion of ribosomal RNA was accomplished by using the NEBNext rRNA Depletion Kit (New England Biolabs). RNA-seq library preparation and analysis was performed as described above.

**Cell stress assays**. Indicated cells lines were grown on 6-well plates until 90% confluency. For positive control of phosphorylation of EIF2AK2 WT and DHX36 KO cells were transfected with 1.5 µg of dsRNA using Lipofectamine 2000 (Thermo Fisher Scientific) and incubated for 6 h. After harvesting, cells were collected and lysed in NP-40 lysis buffer (50 mM HEPES, pH 7.5, 150 mM KCl, 2 mM EDTA, 0.5 mM DTT, 0.5% (v/v) NP-40, protease inhibitors) on ice for 10 min. Lysates were cleared by centrifugation (15 min, 20,000 × $g$). Protein lysates were analyzed by standard western blotting.

**Statistical analyses**. Statistical parameters are shown in the figures and listed in the figure legends. Statistical significance is claimed when $P < 0.05$ in the unpaired or paired two-tailed $t$-tests. In the figures, asterisks mark statistical significance as follows. *$P < 0.05$, **$P < 0.01$, or ***$P < 0.001$.

**Reporting summary**. Further information on research design is available in the Nature Research Reporting Summary linked to this article.

## Data availability

PAR-CLIP, ribosome footprinting, and RNA-seq data have been deposited in the National Center for Biotechnology Information (NCBI) Sequence Read Archive under the accession number GSE105175. All data is available from the authors upon reasonable request.

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

## Acknowledgements

Research in the Paeschke laboratory is funded by the Deutsche Forschungsgemeinschaft (DFG, German Research Foundation) under Germany's Excellence Strategy – EXC2151 – 390873048 as well as an ERC Stg Grant (638988-G4DSB). Work in the Hafner laboratory is supported by the Intramural Research Program of the National Institute for Arthritis and Musculoskeletal and Skin Disease (NIAMS). We thank the NIAMS Genomics Core Facility and Dr. Stefania Dell'Orso and Gustavo Gutierrez-Cruz (NIAMS) for sequencing support and Dr. Suman Ghosal (NIAMS) for binding motif analysis. We thank Dr. Andreas Schlosser (University of Würzburg) for Mass Spectrometry analysis of SILAC experiments and Dr. Martin Schlee for providing dsRNA. We thank the iPSC/CRISPR Centre (UMCG/ERIBA) for support by creating the DHX36 KO cell line, as well as the Flow Cytometry Core Facility and the Microscopy Core Facility (both University Hospital Bonn) for instrumental support. We thank Dr. Alexander Buchberger (University of Würzburg) for generous help and support during the whole period of this project and Maria Gallant and Eike Schwindt for experimental support. We thank Alexis Jacob (NIAMS), Dr. Henrike Maatz (MDC Berlin), and Dr. Satyaprakash Pandey (ERIBA, Groningen) for careful reading of the manuscript.

## Author contributions

Conceptualization, M.H. and K.P.; Methodology, M.S., S.A.J., J.M., D.H., D.B., A.D.M, H.G.K., and X.W.; Data analysis, M.S., S.A.J., J.M., A.D.M, M.H.; Writing—original draft, M.S., M.H., K.P.; Writing—review and editing, M.S., S.A.J., J.M., D.B., M.H. and K.P.; Funding acquisition, M.H. and K.P.; Resources, M.H. and K.P.; Supervision, M.H. and K.P.

## Additional information

**Competing interests:** The authors declare no competing interests.

