## [Peer Review File · Nature Communications]

Reviewers' comments:

Reviewer #1 (Remarks to the Author):

In this manuscript, the authors demonstrate that the loss of the helicase DHX36 in HEK293 cells leads to the accumulation of mRNA containing a potential G-quadruplex (G4) in their untranslated regions. Specifically, they first show that DHX36 is a cytoplasmic helicase by subcellular fractionation, it interacts with mRNA in HEK293 and does not directly interact with ribosomes. After, they did PAR-CLIP to identify the DHX36 RNA recognition element. They then reanalysed their PAR-CLIP data and compared their data with the rG4-seq made previously by Kwok et al, they show that DHX36 binds G-rich sequences that can form rG4s in vitro. After that, a mRNA half-life determination (of genes containing a G4) and ribosome profiling show that loss of DHX36 increases target mRNA stabilisation and that DHX36 increase translational efficiency. Finally, they "briefly" showed that DHX36 loss contribute to the activation of the cellular stress response and hypothesize a mechanism in which DHX36 loss leads to the accumulation of its G-rich mRNA targets in stress granules.

Overall, this manuscript is of excellent quality and supported by many experiments in vitro and in cellulo. The presented findings are relevant to the G-quadruplexes and RNA-binding protein fields. However, some conclusions might be "too strong" considering the results presented. For example, on page 18 L354-356, the authors wrote "Our data suggest that DHX36 loss resulted in increased abundance of rG4s [...] while at the same time rendering them translation incompetent". The results presented previously (Fig. 5) rather show a decrease in translation efficiency, but are still translated, so they are not completely translation incompetent but should be nuanced to "less competent".

The last section of this manuscript, "Stress granule and cellular stress response", is weaker in conclusions and experiments. The authors propose an interesting hypothesis that DHX36 RNA targets correlates with RNA enriched in stress granules identified in a recent work. However, to state that "DHX36-KO [...] leads to the sequestration and stabilization of rG4 transcript in SGs" (P20 L-390-392) seems far-fetched and no conclusive evidence of that is presented in this manuscript. Experiments like using the BG4 antibody to demonstrate co-localization of G4 with SGs, measuring the number and sizes of SGs upon DHX26-KO and possible rescue with the transgenic FH-DHX36 for example could have strengthened that conclusion which seem for now only a possible hypothesis.

Minor comments:

Introduction:

1. References 3 and 4 are reviewed that dated (2012) considering that the field of RNA G4 has significantly progressed. Rouleau et al. (2017) would be more appropriated.

Results section:

1. What is the rationale for selecting the WAC, PURB and SLMO2 mRNAs for the confirmation assays from all the possible DHX36 mRNA targets?
2. Suppl. Fig. 1 c and d should be after e in your supplementary figures
3. You mention that we can observed a "statistically significant decrease in ribosome protected fragments independent of whatever the binding site sites were found in the CDS or the UTRs". Since both you p-value for the 3'UTR and 5'UTR is higher than 0.05, explain how these values can still be significant?
4. At many times you use the term "potentially rG4-forming" (L199, L207, L249, L313, L317) for rG4 detected by rG4-seq. This term is more used for rG4 predicted in silico (as in [54]).
5. For reference 35 (non-canonical G4RNA), the work of Jodoin et al (RNA, 2014) would be more appropriate since the demonstration is in vitro as well as in cellulo.

Online Methods

1. In Western Blot analysis, there is not mention of washes after antibodies incubation
2. In subcellular fractionation, there is not mention of the device used for sonication, power parameter is then useless
3. It is unclear if the Circular dichroism experiments confirming the G-quadruplex folding of some DHX36 targets was performed with DNA or RNA oligonucleotides. Considering that the oligo samples were incubated for 10 minutes at 95°C, I supposed that it was DNA. As the scope of the manuscript is to study RNA targets of DHX36, it might be more relevant to perform this CD experiment with RNA sequences.
4. The section "cell stress assay" presents a protocol including treatment of cells with NaAsO₂, however, no such results are presented anywhere in the manuscript. On the other hand, the method used to verify cellular stress response upon dsRNA addition (fig 7) is not detailed.

Reviewer #2 (Remarks to the Author):

Sauer and collaborators' work provides a better and deeper insight into DHX36's role in the stability and translation of mRNAs related to its ability to link and unfold RG4 structures. This manuscript is likely to have a wider impact on the community working in the RNA field and more specifically on those interested in understanding the role of RG4s and their regulators in post-transcription gene expression. Before publication, a number of issues need to be clarified to fully support the proposed model. The association of DHX36 with translational machinery as well as the contribution of AU-rich elements to translation and mRNA stability requires further analysis to strengthen the main message of the manuscript and the model

29 : we hypothesize that these mRNAs are stabilized by formation of G4s, yet rendered translationally incompetent by sequestration in SGs.

It should be indicated in the absence of DHX36

58 : that neither transient knockdown of specific helicases, nor ATP-depletion have resulted in the..

Misleading : only DHX36 has been tested in ref 8

78 : propose the model that DHX36 loss results in the formation of rG4s and other structures on target

What's the model in non-depleted cells? It would be worth adding a figure depicting the proposed model +/- DHX36

84: The molecular and structural basis for unwinding of G4 structures by DHX36 is well understood

It would be important to provide a more detailed description of the results in ref 25: are they transposable to RNA binding?

Fig 1A is not useful in the body of the manuscript, it can be moved to Suppl

Fig 1d: DHX36 is observed interacting with poly-A+ mRNAs also in the absence of cross-linking. I don't see the point of doing +/- UV

88: known splice isoforms that differ by alternative 5' splice site usage in exon 13 (Fig. 1a).

Is there a difference in function between the two isoforms?

106 remainder migrated with the monosomal and polysomal fractions (Fig. 1e)

I can't see the presence of DHX36 in polysomes. The contrast is quite strong. In the preprint version of this manuscript, we could clearly see the presence of DH36 in the polysome and shift. Unlike the results in Fig. 1e, in the recent work from Murat et al (Genome Biol2018), DHX36 is

present in polysomal fractions and is suggested to function as a direct translational regulator. Is the association to polysomes cell-type specific? It would be important to consider these results and discuss them. Also, it would be important to add as control an initiation factor, such as eIF4A. Is the distribution similar to DHX36? If so, the conclusion "arguing against a role in translation" should be reviewed and DHX36 might be proposed as a translation initiation regulator.

159 analysis revealed an enrichment of FH-DHX36-E335A binding sites within the first 100 nt of the

Is there an explanation for this enrichment?

193 In the following, we will focus our functional analysis on mRNA targets obtained by FH-DHX36-E335A PAR-CLIP, considering its high correlation with the FH-DHX36 PAR-CLIP

I agree, but this mutant binds also AU-rich elements known to regulate the stability of mRNAs. I find that this information is not considered throughout the manuscript.

421 Nevertheless, a DHX36 mutant with an inactive helicase (DHX36-E335A) domain also crosslinked at AU-rich sequences, suggesting that these sites may serve as additional recruitment platforms, but that the active protein rapidly translocates to the more structured G-rich regions being unwound.

Are AU-rich elements located in the vicinity of the rG4s?

316 selected target sites and generated reporter cells stably expressing a GFP control and an mCherry-coding transgene fused to potentially G4-forming DHX36 PAR-CLIP binding sites on the WAC and PURB mRNA

Can this effect be reverted by mutating the G4 sequence?

320 Taken together with our observation that >90% of DHX36 proteins did not co-sediment with translating ribosomes (Fig. 1e) and thus unlikely influenced translation elongation, our data suggest that DHX36 increased the translational competence of mRNAs, possibly allowing access to the translational machinery either by resolving rG4s blocking translation initiation or by changing localization of target mRNAs.

I am not comfortable with this sentence since I am not completely convinced of the results of Fig 1e.

348 cytoplasm increased by ~1.5 fold, indicating an accumulation of rG4s upon DHX36-KO that corresponded in magnitude with the levels of DHX36 target mRNA stabilization

Does this result mean that the increase in the BG4 signal results from the increasing of mRNA levels after DHX36 silencing? Given that the amount of mRNAs containing G4 varies according to the presence of DHX36, it is difficult to conclude on the accumulation of structured rG4s upon DHX36 silencing.

Fig. 5i and

312 We found that potentially rG4-forming RNAs exhibited a 31317% decreased TE upon DHX36-KO (Fig.5i), further supporting a role of DHX36 in resolving rG4s.

What's the contribution of the AU-rich elements to TE? And to mRNA stability (Fig. 4d)?

Fig. 6 It would be important to control the loss of cytoplasmic BG4 staining after RNase A treatment and to perform IF with the DHX36 antibody. Does DHX36 colocalize with SGs or PB? Does DHX36 silencing induce SG/PB formation? BG4 staining and colocalization with SGs after cPDS addition would strengthen the conclusion and support the proposed model

380 DHX36 itself was found to localize to SGs and thus, we cross-referenced our PAR-CLIP data with a recently published dataset of transcripts enriched in SGs.

The fact that both the DHX36 and its targets are found in SGs is not compatible with the proposed model in which DHX36 targets are redirected to the SGs following DHX36 silencing

We thank both referees for their nice and constructive comments.

Below please find our responses (underlined and in italics) to the reviewers' critiques.

Reviewer #1:

In this manuscript, the authors demonstrate that the loss of the helicase DHX36 in HEK293 cells leads to the accumulation of mRNA containing a potential G-quadruplex (G4) in their untranslated regions. Specifically, they first show that DHX36 is a cytoplasmic helicase by subcellular fractionation, it interacts with mRNA in HEK293 and does not directly interact with ribosomes. After, they did PAR-CLIP to identify the DHX36 RNA recognition element. They then reanalysed their PAR-CLIP data and compared their data with the RG4-seq made previously by Kwok et al, they show that DHX36 binds G-rich sequences that can form rG4s in vitro. After that, a mRNA half-life determination (of genes containing a G4) and ribosome profiling show that loss of DHX36 increases target mRNA stabilisation and that DHX36 increase translational efficiency. Finally, they “briefly” showed that DHX36 loss contribute to the activation of the cellular stress response and hypothesize a mechanism in which DHX36 loss leads to the accumulation of its G-rich mRNA targets in stress granules.

Overall, this manuscript is of excellent quality and supported by many experiments in vitro and in cellulo. The presented findings are relevant to the G-quadruplexes and RNA-binding protein fields. However, some conclusions might be “too strong” considering the results presented. For example, on page 18 L354-356, the authors wrote “Our data suggest that DHX36 loss resulted in increased abundance of rG4s [...] while at the same time rendering them translation incompetent”. The results presented previously (Fig. 5) rather show a decrease in translation efficiency, but are still translated, so they are not completely translation incompetent but should be nuanced to “less competent”.

We thank the reviewer for the very positive evaluation of our work („this manuscript is of excellent quality...”) and we are pleased that she/he agreed with us on the high relevance of this work for both the G4 and the RBP fields.

We carefully re-evaluated the manuscript to avoid overinterpretation of the data. For the example the reviewer mentions (page 18, l354-356, now page 19, lines 366-368), we changed the sentence to: “Taken together, our data suggest that DHX36 loss resulted in increased abundance of rG4s and that the formation of these structures stabilized the RNA without corresponding increase in translation”.

The last section of this manuscript, “Stress granule and cellular stress response”, is weaker in conclusions and experiments. The authors propose an interesting hypothesis that DHX36 RNA targets correlates with RNA enriched in stress granules identified in a recent work. However, to state that “DHX36-KO [...] leads to the sequestration and stabilization or rG4 transcript in SGs”(P20 L-390-392) seems far-fetched and no conclusive evidence of that is presented in this

manuscript. Experiments like using the BG4 antibody to demonstrate co-localization of G4 with SGs, measuring the number and sizes of SGs upon DHX26-KO and possible rescue with the transgenic FH-DHX36 for example could have strengthened that conclusion which seem for now only a possible hypothesis.

As the reviewer observed, the last section of our manuscript provides more of an outlook describing promising lines of research to build upon based on our thorough analysis of DHX36 targets and impact on posttranscriptional gene regulation.

Nevertheless, in order to further strengthen the connection between DHX36, RNA G-quadruplexes (rG4), and the integrated stress response, we performed a set of additional experiments: 1.) We analyzed SG formation in WT cell and DHX36 KO cells, as suggested by the referee and include the results in Figure 7. In WT very little cells containing SG can be detected, in DHX36-KO at least 33% of the cells show SG. The rescue cell line expressing transgenic DHX36 in the DHX36-KO background showed similar levels of SG as WT cells.

2.) We tested if elevated PKR levels are the cause for enhanced PKR phosphorylation. For this we probed Western blot with PKR antibody that recognizes the unphosphorylated form. No changes were detectable (Suppl. Fig. S9), indicating that PKR itself is not elevated and further cementing that PKR phosphorylation is a sign of stress.

The new results are in-cooperated in the manuscript and the statement about SG formation are altered according to the new data. We speculate that accumulating mRNAs with G4 and other structures, activate PKR phosphorylation which results in seeding of SG, which in turn downregulates translation of these targets.

Minor comments:

Introduction:

1. References 3 and 4 are reviewed that dated (2012) considering that the field of RNA G4 has significantly progressed. Rouleau et al. (2017) would be more appropriated.

We added two additional, more recent reviews, including the suggested one by Rouleau et al.

Results section:

1. What is the rationale for selecting the WAC, PURB and SLMO2 mRNAs for the confirmation assays from all the possible DHX36 mRNA targets?

We now describe the rationale for choosing targets for validation in the text (p. 13, lines 267-269). We selected sequences that 1.) were found on mRNAs that ranked among the top 100 targets, and 2.) that contained G-rich target sites in the 3'UTR, and 3.) that were found on mRNAs accumulating upon DHX36 KO.

2. Suppl. Fig. 1 c and d should be after e in your supplementary figures

We changed the order of the panels in Suppl. Fig. S1, which is now Suppl. Fig. S2, according to the reviewer's suggestion.

3. You mention that we can observed a “statistically significant decrease in ribosome protected fragments independent of whatever the binding site sites were found in the CDS or the UTRs”. Since both you p-value for the 3’UTR and 5’UTR is higher than 0.05, explain how these values can still be significant?

We thank the reviewer for pointing out our incorrect statement. In fact, the marginal change in RPFs was only significant when ranking our DHX36 binding sites by CDS binding and we rephrased the sentence on p.16 accordingly. It now reads: "Surprisingly, DHX36 loss resulted only in a marginal, albeit statistically significant decrease in ribosome protected fragments (RPFs) ($p < 10^{-5}$, two-sided Kolmogorov-Smirnov (KS) test, Fig. 5a, e and Suppl. Fig. S5a, c, d), particularly for sites in the CDS (Fig. 5b, c and Suppl. Fig. S5b, e, f)." This could hint at a minor role for DHX36 in directly resolving G-rich structures in CDS to clear the way for the ribosome. However, compared to e.g. CNBP (Benhalevy et al., Cell Rep, 2017) this effect is negligible and overshadowed by DHX36 effect on translational efficiency of targets bound in the 3'UTR (Figure 5).

4. At many times you use the term “potentially rG4-forming” (L199, L207, L249, L313, L317) for rG4 detected by rG4-seq. This term is more used for rG4 predicted in silico (as in [54]).

We thank the reviewer for this advice and we changed the mentioned according to her/his suggestion.

5. For reference 35 (non-canonical G4RNA), the work of Jodoin et al (RNA, 2014) would be more appropriate since the demonstration is in vitro as well as in cellulo.

Once again, we are grateful that the reviewer provided us with another excellent reference that we included in the manuscript.

Online Methods

1. In Western Blot analysis, there is not mention of washes after antibodies incubation

In the revised manuscript we described our standard Western blot protocol in greater detail and included information about the washes.

2. In subcellular fractionation, there is not mention of the device used for sonication, power parameter is then useless

We agree with the reviewer and added details about the sonication device used.

3. It is unclear if the Circular dichroism experiments confirming the G-quadruplex folding of some DHX36 targets was performed with DNA or RNA oligonucleotides. Considering that the oligo samples were incubated for 10 minutes at 95°C, I supposed that it was DNA. As the scope of the manuscript is to study RNA targets of DHX36, it might be more relevant to perform this CD experiment with RNA sequences.

The reviewer is correct in her/his assumption that we used DNA oligonucleotides. However, we think that the critical value of our analysis is that the sequence elements we identified in living cells can form parallel G4 in vitro. It is well established that the G4 specificity of DHX36 is determined by the differences between parallel and antiparallel G4s, rather than the backbone of the target molecule (RNA or DNA) as described by Chen and colleagues (Chen et al., Nature, 2018). Furthermore, RNA G4s are even more stable than DNA G4s of the same sequence (Zhang et al., Biochemistry, 2010), further strengthening our case that many of the sequences DHX36 binds to can form parallel G4 structures.

4. The section “cell stress assay” presents a protocol including treatment of cells with NaAsO₂, however, no such results are presented anywhere in the manuscript. On the other hand, the method used to verify cellular stress response upon dsRNA addition (fig 7) is not detailed.

We apologize for this oversight and we now describe the correct assays in the revised methods section.

Reviewer #2 (Remarks to the Author):

Sauer and collaborators' work provides a better and deeper insight into DHX36's role in the stability and translation of mRNAs related to its ability to link and unfold RG4 structures. This manuscript is likely to have a wider impact on the community working in the RNA field and more specifically on those interested in understanding the role of RG4s and their regulators in post-transcription gene expression. Before publication, a number of issues need to be clarified to fully support the proposed model. The association of DHX36 with translational machinery as well as the contribution of AU-rich elements to translation and mRNA stability requires further analysis to strengthen the main message of the manuscript and the model

We thank Reviewer #2 for the positive feedback and her/his prediction of our study's deep impact on the RNA field. We hope that this revised manuscript will further clarify the few open questions the reviewer asked.

29 : we hypothesize that these mRNAs are stabilized by formation of G4s, yet rendered translationally incompetent by sequestration in SGs. It should be indicated in the absence of DHX36

We clarified the sentence and included new results. It now reads: "Considering that DHX36 targets, harboring G4s, preferentially localize in stress granules, and that DHX36 KO resulted in increased SG formation and protein kinase R (PKR/EIF2AK2) phosphorylation, we speculate that DHX36 is involved in resolution of rG4 induced cellular stress".

58 : that neither transient knockdown of specific helicases, nor ATP-depletion have resulted in the. Misleading : only DHX36 has been tested in ref 8

We agree with the referee and changed the statement to read: "A large level of redundancy of the rG4-interacting machinery is expected, considering that neither transient knockdown of a helicase unwinding rG4s in vitro, DHX36, nor ATP-depletion have resulted in the increase of rG4 formation above the threshold that could be reliably detected by dimethyl sulfate sequencing".

78 : propose the model that DHX36 loss results in the formation of rG4s and other structures on target What's the model in non-depleted cells? It would be worth adding a figure depicting the proposed model +/- DHX36

We added a model as Fig. 8.

84: The molecular and structural basis for unwinding of G4 structures by DHX36 is well understood. It would be important to provide a more detailed description of the results in ref 25: are they transposable to RNA binding?

The structural analysis by Chen et al (Nature, 2018) indicates that G4 specificity of DHX36 is determined in the first place by the differences between parallel and antiparallel G4s and not by the backbone of the target molecule (RNA or DNA), considering that they do not observe interactions between DHX36 and the nucleotide backbone. Furthermore, as observed by others, DHX36 can efficiently unwind DNA, as well as RNA G4 that almost exclusively fold into parallel G4 (Xiao et al., Sci Rep, 2017), and its cytoplasmic localization indicates that it will likely - at least in HEK293 - exclusively interact with RNA. Finally, RNA G4s are even more stable than DNA G4s of the same sequence (Zhang et al., Biochemistry, 2010), further strengthening our case that many of the sequences DHX36 binds to can form G4 structures

We added more detail on DHX36 activity on RNA and DNA G4 to page 5 ("It is clear that the N-terminal domain together with an OB-fold of DHX36 specifically recognizes parallel DNA and RNA G4 and unfolds G4 in an ATP-dependent manner". Furthermore, we included a sentence in the discussion pointing out that rG4 mainly folds into parallel G4, which is in agreement with DHX36 binding and unfolding preferences (page 24: "Finally, DHX36 prefers to unwind parallel G4 structures, exactly the kind typically formed on RNA, further supporting its posttranscriptional regulatory role in the cytoplasm").

Fig 1A is not useful in the body of the manuscript, it can be moved to Suppl

See also above critique by Reviewer #1. We moved the schematic to Suppl. Fig. S1a.

Fig 1d: DHX36 is observed interacting with poly-A+ mRNAs also in the absence of cross-linking. I don't see the point of doing +/- UV

We agree with the reviewer that this information is not essential and to avoid any confusion, we removed the -UV panel from what is now Fig. 1c.

88: known splice isoforms that differ by alternative 5' splice site usage in exon 13 (Fig. 1a). Is there a difference in function between the two isoforms?

Reviewer #2 raises an important point. Besides the initial report describing DHX36 (Tran et al., Mol Cell, 2004) none of the published research addressed differences between both isoforms. Tran et al. suggested a differential localization of both isoforms, a result which we cannot confirm (see Fig. 1c, new Fig. 1b). In none of our analysis did we detect differences between both isoforms. Therefore, we consider their function as redundant and we added the following sentence to the results section on page 5: "Because no changes in the two isoforms could be detected (Fig. 1b), we did not further discriminate between them."

106 remainder migrated with the monosomal and polysomal fractions (Fig. 1e)

I can't see the presence of DHX36 in polysomes. The contrast is quite strong. In the preprint version of this manuscript, we could clearly see the presence of DH36 in the polysome and shift. Unlike the results in Fig. 1e, in the recent work from Murat et al (Genome Biol2018), DHX36 is present in polysomal fractions and is suggested to function as a direct translational regulator. Is the association to polysomes cell-type specific? It would be important to consider these results and discuss them. Also, it would be important to add as control an initiation factor, such as eif4A. Is the distribution similar to DHX36? if so, the conclusion "arguing against a role in translation" should be reviewed and DHX36 might be proposed as a translation initiation regulator.

Please note that we used exactly the same figure panel for the preprint and current version of the manuscript. The quality differences are likely due to downsampling of our figure by Nature Communication's manuscript submission system.

We agree with the referee that our results contrast with the recent published polysome gradients from Murat et al. that show a stronger association to polysomes. We repeated our polysome gradient experiments multiple times, including in the FH-DHX36 cell line overexpressing DHX36 fourfold compared to endogenous levels and we reproducibly observe the vast majority of DHX36 in the soluble fraction (Fig. 1d and Suppl. Fig. S1b). Differences to Murat et al. either reflect the use of a different cell

line, or the use of different reagents. In our case, we validated the specificity of our antibody using the DHX36-KO cell line.

We followed the reviewer's advice and also analyzed EIF4A migration on the polysome, which indeed shows a similar pattern as DHX36 (Fig. 1e, new Fig. 1d). We rephrased our conclusion from these experiments a bit more cautiously to reflect that at this stage we cannot exclude a function of DHX36 in translational initiation (page 6: "Considering that the translation initiation factor EIF4A shows a similar distribution on polysomes, our data indicate that DHX36 does not affect translation elongation, but could possibly be involved in translation initiation." and page 17: "Taken together with our observation that >90% of DHX36 proteins did not co-sediment with translating ribosomes (Fig. 1d) and thus unlikely influenced translation elongation, our data suggest that DHX36 increased the translational competence of mRNAs, possibly allowing access to the translational machinery either by resolving rG4s blocking translation initiation or by changing localization of target mRNAs."). We further acknowledge that translational initiation may be impaired by DHX36-KO in the discussion on page 25: "Thus, the stabilized G-rich target mRNA in DHX36-KO cells were not translational competent, either due to decreased translation initiation, or by sequestration of these RNAs into granules, such as P-bodies or SGs."

159 analysis revealed an enrichment of FH-DHX36-E335A binding sites within the first 100 nt of the

Is there an explanation for this enrichment?

The enrichment of DHX36 binding sites within the first 100 nt of CDS mirrors those of other G-rich binding proteins (e.g. CNBP, Benhalevy, Cell Rep, 2017). It is however not predictive of DHX36 regulatory impact and likely rather reflects sequence composition or other binding preferences of DHX36.

193 In the following, we will focus our functional analysis on mRNA targets obtained by FH-DHX36-E335A PAR-CLIP, considering its high correlation with the FH-DHX36 PAR-CLIP I agree, but this mutant binds also AU-rich elements known to regulate the stability of mRNAs. I find that this information is not considered throughout the manuscript.

We agree with the reviewer that a deeper look at the AU-rich regions found in the DHX36E335A PAR-CLIP is warranted. Please note that AU-rich in this case does not mean a traditional AU-rich element that is characterized by the presence of repeats of the AUUUA pentamer; in our case it refers to pentamers consisting completely of A and U and containing at least one A and U.

In order to understand the contribution of AU-rich target sites to DHX36-mediated gene regulation, we used our RNAseq data from WT and DHX36-KO cells and binned our DHX36 target mRNAs by whether binding sites contained AU-rich clusters or G-rich clusters. Note that due to the depth of our dataset it was impossible to find sufficient target mRNAs to perform this analysis that contained exclusively AU-rich or G-rich clusters. Thus, we did the next best thing and ranked our targets

according to whether they contained at least five of (1.) G-rich clusters, (2.) AU-rich clusters, or (3.) clusters containing both AU-rich and G-rich pentamers. Virtually all of these mRNAs contain additional clusters from the other categories. Figure 1 below documents that all of these targets are regulated, but those with AU-rich clusters are accumulating to higher levels upon DHX36-KO. This is consistent with our speculation that AU-rich elements with little intrinsic structure support helicase structure, e.g by serving as an additional landing platform for the helicase from which it translocates to find and unwind G-rich structures. Considering that this analysis is not ideal due to the lack of targets with exclusive G-rich or AU-rich DHX36 binding sites, it is not fully conclusive and we would leave it to the discretion of the reviewer and the editor whether to include in the main manuscript.

Figure 1: Cumulative distribution analysis of mRNA abundance changes comparing wt HEK293 and DHX36-KO cells. DHX36 targets were binned such that they contained more than 5 PAR-CLIP sites in each bin and ranked by number of sites containing clusters that were G-rich (red line), AU-rich (orange line), or mixed containing AU-rich and G-rich pentamers in the same cluster, (brown line).

431 Nevertheless, a DHX36 mutant with an inactive helicase (DHX36-E335A) domain also crosslinked at AU-rich sequences, suggesting that these sites may serve as additional recruitment platforms, but that the active protein rapidly translocates to the more structured G-rich regions being unwound. Are AU-rich elements located in the vicinity of the RG4s?

See response to previous comment. Virtually all mRNAs with DHX36 binding sites contain sites with AU- and G-rich clusters in the 3'UTR. We mention these observations on pages 13 and 23.

316 selected target sites and generated reporter cells stably expressing a GFP control and an mCherry-coding transgene fused to potentially G4-forming DHX36 PAR-CLIP binding sites on the WAC and PURB mRNA. Can this effect be reverted by mutating the G4 sequence?

To address this question, we created a set of new reporter cell lines that contained mutations in the G4-forming region of WAC and PURB (Suppl. Fig. S7). Whereas reporter protein levels from wildtype cluster plasmids are increased in WT cells (Ratio WT/KO >1), mutation of the G4-forming motif in those clusters showed no differences in the reporter protein level between WT and DHX36-KO cells (Ratio WT/KO ~1). This is also the case for reporter plasmids containing non-DHX36 binding sites of the DDX5 mRNA as determined by PAR-CLIP (Ratio WT/KO ~ 1).

320 Taken together with our observation that >90% of DHX36 proteins did not co-sediment with translating ribosomes (Fig. 1e) and thus unlikely influenced translation elongation, our data suggest that DHX36 increased the translational competence of mRNAs, possibly allowing access to the translational machinery either by resolving rG4s blocking translation initiation or by changing localization of target mRNAs.

I am not comfortable with this sentence since I am not completely convinced of the results of Fig 1e.

Please see our response above. We have now weakened our statements regarding DHX36 role in translation to reflect that we cannot formally rule out a role in translation initiation at this stage, though it appears difficult to reconcile with the fact that the functional DHX36 binding sites we found resided in the 3'UTR of target mRNAs.

348 cytoplasm increased by ~1.5 fold, indicating an accumulation of rG4s upon DHX36-KO that corresponded in magnitude with the levels of DHX36 target mRNA stabilization Does this result mean that the increase in the BG4 signal results from the increasing of mRNA levels after DHX36 silencing? Given that the amount of mRNAs containing G4 varies according to the presence of DHX36, it is difficult to conclude on the accumulation of structured RG4s upon DHX36 silencing.

The BG4 antibody was extensively characterized by the Balasubramanian group (Biffi et al., Nat Chem, 2013 and 2014) as specific for G4 and is thus unlikely to recognize unstructured G4, even if the RNA had the potential to form those. Thus, a mere increase in abundance of these RNAs, without formation of some G4 structures is unlikely to result in an increase of BG4 signal. As an additional control we now also include BG4 immunofluorescence documenting an increase in BG4 signal upon cPDS treatment (Suppl. Fig. S8).

Fig. 5i and 312 We found that potentially rG4-forming RNAs exhibited a 17% decreased TE upon DHX36-KO (Fig.5i), further supporting a role of DHX36 in resolving rG4s. What's the contribution of the AU-rich elements to TE? And to mRNA stability (Fig. 4d)?

Please see our response to the point about the AU-rich RNAs above.

Fig. 6 It would be important to control the loss of cytoplasmic BG4 staining after RNase A treatment and to perform IF with the DHX36 antibody. Does DHX36 colocalize with SGs or PB? Does DHX36 silencing induce SG/PB formation? BG4 staining and colocalization with SGs after cPDS addition would strengthen the conclusion and support the proposed model

We performed the requested experiments and added them to Fig. 6 and Suppl. Fig. S8. As expected, cPDS treatment increases BG4 signal and RNase A treatment abrogated it. Considering that the BG4 signal was more diffuse than the SG foci, we cannot show exclusive colocalization. Nevertheless, it is established that SG formation is preceded by seeding of submicroscopic granules (Protter and Parker, Trends Cell Biol, 2016) and that we do not observe a full-blown stress response with shutdown of translation that usually results in localization of virtually all polyA RNA in SGs, we do not expect to observe complete colocalization of BG4 and SG signal.

380 DHX36 itself was found to localize to SGs and thus, we cross-referenced our PAR-CLIP data with a recently published dataset of transcripts enriched in SGs. The fact that both the DHX36 and its targets are found in SGs is not compatible with the proposed model in which DHX36 targets are redirected to the SGs following DHX36 silencing

We do not agree with the reviewer on this point. Our data indicate that 1.) DHX36 targets preferentially localize to SGs, 2.) G4 containing RNAs preferentially localize to SGs, and 3.) that DHX36 silencing results in induction of the stress response. Our model (Fig. 8) predicts that DHX36 prevents mRNA from aggregating in SGs, and since WT cells do not show signs of stress this would likely happen before SG formation. The fact that DHX36 itself does localize to SGs upon arsenite stress is likely secondary, as upon stress its targets move to SG (Chalupnikova et al., J Biol Chem, 2008).

REVIEWERS' COMMENTS:

Reviewer #1 (Remarks to the Author):

I'm satisfied with the revised version. Nice job. Congratulations!

Reviewer #2 (Remarks to the Author):

I am satisfied with all the answers from the authors and with the changes made. I'd only make a modification of figure 8 so that it may better fit with "DHX36 loss resulted in a concomitant change in translation". Translation regulation does not appear in the drawn model

Point by point response:

We thank the referee for their time and positive evaluation of our manuscript.

REVIEWERS' COMMENTS:

Reviewer #1 (Remarks to the Author):

I'm satisfied with the revised version. Nice job. Congratulations!

We thank the referee for this positive feedback and his/her support in the last round of revision.

Reviewer #2 (Remarks to the Author):

I am satisfied with all the answers from the authors and with the changes made. I'd only make a modification of figure 8 so that it may better fit with "DHX36 loss resulted in a concomitant change in translation". Translation regulation does not appear in the drawn model

We agree that the Figure 8 lacked the translational aspect of our story and we modified the figure according to his/her suggestions. We also thank him/her for his time and constructive, fair review process.